# Global, regional, and national burden of hypertensive intracerebral hemorrhage, 1990 to 2021 and projections to 2050: Results from the Global Burden of Disease Study 2021

Chao Zhang[1o], Jiao Chen[2o], Linjing Song[1], Juwei Dong[1], Qianqian Hu[3], Dayong Ma[1‡*], Jing Li[4‡*]

1 Department of Encephalopathy, Dongzhimen Hospital, Beijing University of Chinese Medicine, Beijing, China, 2 Experimental Research Center, Chinese Academy of Chinese Medicine, Beijing, China, 3 Department of Cardiovascular, Gansu University of Chinese Medicine, Lanzhou, Gansu, China, 4 Department of Endocrine, Beijing Hospital of Chinese Medicine, Capital Medical University, Beijing, China

⊙ These authors contributed equally to this work.
‡ DM and JL also contributed equally to this work.
* a3153@bucm.edu.cn (DM); lijing11442@bjzhongyi.com (JL)

## Abstract

### Objective

Hypertensive intracerebral hemorrhage (HICH) is characterized by high morbidity, mortality, disability, and recurrence. According to the current study, there have been no targeted studies exploring the epidemiology and trends of HICH since the Global Burden of Disease (GBD) study report in 2021. The aim of this study was to assess deaths and disability-adjusted life years (DALYs) of patients with HICH globally from 1990 to 2021, with projections to 2050.

### Methods

We analyzed age-standardized death rates (ASDR) and age-standardized DALYs for HICH across various countries, geographic regions, age groups, sexes, and the sociodemographic index (SDI) using data from the 2021 GBD study. In addition, we used a Bayesian age-period-cohort (BAPC) model to project the burden of HICH from 2021 to 2050.

### Results

In 2021, the ASDR for HICH was 22.641 per 100,000 people, reflecting a 31.418% decrease from 1990. Similarly, the age-standardized DALYs rate was 521.085 per 100,000 people, marking a 32.163% reduction since 1990. Country and regional patterns showed stark contrasts: Nauru and Mozambique had the highest ASDRs and age-standardized DALY rates, while Switzerland and Canada reported the lowest.

**Data availability statement:** All data are available from the Global Health Data Exchange query tool (GHDx) (http://ghdx.healthdata.org/gbd-results-tool).

**Funding:** This research was supported by the National Natural Science Foundation of China (Grant No.: 82274458, Recipient: Dayong Ma).

**Competing interests:** The authors have declared that no competing interests exist.

Regionally, Central Africa, South Africa, Central Asia, East Asia, and Southeast Asia had the highest rates, whereas the Americas, Europe, and Oceania had the lowest. Age and gender trends indicated global peaks in the ASDRs (90–94 age group) and age-standardized DALY rates (85–89 age group), with men having higher rates across all age groups. Additionally, both ASDRs and age-standardized DALY rates were negatively associated with SDI levels. Projections from 2021 to 2050 suggest a continued overall decline in global ASDRs and age-standardized DALYs rates for HICH.

## Conclusion

From 1990 to 2021, and projected from 2021 to 2050, the global ASDR and age-standardized DALYs rate for HICH have shown an overall decline. However, significant disparities persist between countries and regions, with less developed areas facing a disproportionately higher burden. In these regions, the early implementation of targeted prevention and treatment strategies, alongside continuous improvements in healthcare resources and services, is crucial to mitigating the global burden of HICH.

## Introduction

Intracerebral hemorrhage (ICH) is the most devastating form of stroke [1], contributing significantly to global morbidity and mortality [2], and imposing a substantial burden on healthcare systems worldwide [3].Hypertensive intracerebral hemorrhage (HICH) is the predominant subtype of ICH, is strongly associated with hypertension, and accounts for approximately 70% of all ICH cases [4]. Advances in detection methods, improved treatments, and heightened awareness have influenced the morbidity and mortality rates of ICH [5].A study based on computed tomography found that noncontrast CT has a sensitivity of up to 95% in the detection of ICH and has been widely used [6]. Studies on the treatment of ICH have found that blood pressure control trials reduce ICH mortality by 10–15% [7], after administration of anticoagulant-specific reversal agents (e.g., idarucizumab, prothrombin complex concentrate), mortality and hematoma enlargement rates in patients with ICH are reduced by approximately 30%−40% compared to baseline levels without reversal agents or conventional therapy [8]. Public health campaigns focusing on hypertension control increased regular blood pressure checks by 20–30% in some regions and reduced smoking rates by 10–15% in certain communities, thereby lowering ICH risk [9]. In addition, a stroke study based on the GBD database from 1990 to 2019 found a 17.0% decrease in incidence, a 60% decrease in prevalence, a 36.0% decrease in mortality and disability-adjusted life years (DALYs) for ICH compared with ischemic stroke [10]. Despite improvements in clinical management, high systolic blood pressure, a major risk factor for HICH, continues to be strongly associated with factors such as smoking, diet, and the environment, and varies by country, region, gender, and SDI level [11]. In 2021, high systolic blood pressure was a major contributor to high ASDRs and age-standardized DALYs. Other

important risk factors include particulate matter pollution, smoking, indoor air pollution from solid fuels, and high sodium diets, and there are gender differences in these risks [12]. Regionally, Central and Southeast Asia had a high prevalence of hypertension, exacerbated by unhealthy habits like high-salt diets and smoking, thus increasing the ICH burden [13]. In low SDI areas, household air pollution from solid fuels accounted for 40.2% of all ICH deaths, compared to only 8.1% and 5.9% for particulate matter pollution and smoking, respectively. In high SDI regions, deaths from these pollutants were lower [12].Thus, HICH has emerged as a major public health problem as a leading cause of death and long-term disability among adults worldwide, especially in low- and middle-income countries [14–16].Therefore, there is a need to explore why differences in spatial and temporal trends, gender, and age exist in HICH—for example, "in the Republic of Nauru, patients have low income levels and do not have access to standardized treatments [17], which can complicate and make hospitalization unpredictable; in the Republic of Mozambique, there is a severe shortage of health workers [18], which can prevent timely treatment and prolong hospitalization"—to guide prevention and treatment efforts.

The GBD study offers a comprehensive framework for evaluating the impact of ICH across various regions and over time [19]. Although most previous studies related to GBD [12] have focused on analyzing risk factors and distributional differences in ICH, this paper builds on previous studies by providing a detailed analysis of hypertension as an important risk factor and systematically analyzes the impact of geographic location, SDI, age, and sex differences on trends in ASDRs and age-standardized DALYs. To our knowledge, this GBD-based study is one of the most comprehensive studies to date analyzing the global burden of HICH in terms of mortality, DALYs, and projected future trends. These findings may inform public health interventions in specific regions, by gender, at different ages, etc., and can also help improve discharge protocols or resource allocation.

## Materials and methods

### Ethical statement

The data utilized in this study were derived from publicly available retrospective data from the GBD study. For the purpose of data extraction and collation in this manuscript, the specific operational period for accessing and organizing these GBD data was from February 5, 2025, to February 16, 2025, and the study was approved by the Medical Ethics Committee of Dongzhimen Hospital, Beijing University of Chinese Medicine (2024DZMEC-205–01), which concluded that the requirement for subjects to provide informed consent was waived because the study used publicly available data and did not include any personally identifiable information. This study followed the Guidelines for Accurate and Transparent Reporting of Health Estimates for Cross-sectional Studies (GATHER) (https://www.who.int/publications/m/item/gather-checklist) [20]. HICH is usually a spontaneous intracerebral hemorrhage that occurs in patients with a history of chronic hypertension (blood pressure ≥140/90 mmHg or a previous diagnosis of hypertension) and excludes cases caused by other etiologies (e.g., cerebral aneurysms, arteriovenous malformations, trauma, or coagulation disorders) [21].

### Data sources

This study is part of the GBD 2021 update, which estimates epidemiological information for 204 countries and territories and provides comparative analyses of 369 diseases and injuries [22]. We obtained data on the burden of HICH from 1990 to 2021 from the Global Health Data Exchange query tool (GHDx) (http://ghdx.healthdata.org/gbd-results-tool). The data from the GBD 2021 study were processed and analyzed as follows: Data were statistically analyzed and visualized using R 4.4.1 software. A linear regression model was used to calculate the EAPC and its 95% CI to assess the temporal trends of age-standardized rates (ASRs). A BAPC model was used to approximate marginal posterior distributions using the Integrated Nested Laplace Approximation (INLA) method to project ASDRs and age-standardized DALYs rates for HICH from 2025 to 2050 [23,24]. Using the GBD 2021 dataset, we extracted the mortality rates, DALYs, and their corresponding ASRs for HICH for the period 1990–2021.

## The Sociodemographic Index (SDI)

The SDI is a composite indicator of socio-demographic development, ranging from 0 to 1, with values closer to 1 indicating better socio-economic development [25]. In this study, we categorized countries and regions into five SDI categories: low SDI (0–0.454743), low-medium SDI (0.454743–0.607679), medium SDI (0.607679–0.689504), high-medium SDI (0.689504–0.805129), and high SDI (0.805129–1), to examine the relationship between the burden of HICH and socio-economic development. The SDI data used in this study came from the Institute for Health Metrics and Evaluation (IHME) [26]. SDI stratification is critical to HICH research. It takes into account differences in health care access, prevention, and management across SDI levels and facilitates targeted analysis of socio-economic influences on HICH trends, thus helping to target interventions to specific regions.

## Projection analysis

A Bayesian age-period-cohort (BAPC) model was employed to predict the ASDR, the DALYs, and the age-standardized DALYs rate for HICH from 2025 to 2050. BAPC modeling is a methodology used in epidemiology and biostatistics to analyze the relationship between incidence rates and time. It uses sample data and a priori information to obtain unique parameter estimates [27], allows for the inclusion of known risk factors as covariates in the model, and also simulates the impact of future changes in healthcare by setting up different scenarios. Based on the assumption that the effects of age, period, and cohort are similar in time, the Bayesian inference in the BAPC model utilizes second-order stochastic bias to smooth the three aforementioned prior values and predict the posterior rate [12]. BAPC employs the Integrated Nested Laplace Approximation (INLA) to approximate the marginal posterior distributions, thereby avoiding the mixing and convergence problems associated with the Markov Chain Monte Carlo method and the traditional Bayesian approach, which has been widely used to analyze trends in chronic diseases and predict future disease burden [28].

## Statistical analysis

All statistical analyses and data visualizations were conducted using R (version 4.4.1). Descriptive statistics were performed for all key variables, with results expressed as means and their 95%UI, estimated annual percentage change (EAPC), and their 95% confidence intervals (CI). The EAPC is a commonly used metric in epidemiological studies to assess trends in age-standardized ratios (ASRs) of diseases over time. The coefficient $\beta$ is derived from the natural logarithm of the ASRs, where $y$ represents $In$(ASR) and $x$ represents calendar years. The EAPC and its 95% CI were determined by the following linear regression model [29]:

$$y = \alpha + \beta x + \varepsilon EAPC = 100 * (\exp(\beta) - 1)$$

When the lower limit of the 95% CI exceeds 0, it indicates an upward trend; when the upper limit is below 0, it indicates a downward trend. If the 95% CI contains 0, it indicates no statistically significant change in the trend pattern. In trend analysis, $P < 0.05$ is considered statistically significant.

## Results

### Global estimates of the burden of death and DALYs from HICH

Tables 1 and 2 summarize the global burden of ASDRs and age-standardized DALYs rates for HICH. As shown in Tables 1 and 2, the global ASDR for HICH in 2021 was 22.641 per 100,000 people, a decrease of 31.418% from 1990 (Table 1), and the age-standardized DALYs rate was 521.085 per 100,000 people, a decrease of 32.163% since 1990 (Table 2). ASDR (EAPC = −1.369; 95% CI:-1.550--1.187) (Table 1) and age-standardized DALYs rate (EAPC = −1.401; 95% CI:-1.568--1.234) (Table 2) for HICH trended downward globally between 1990 and 2021.The decreasing trend of ASDR

**Table 1. ASDR of HICH between 1990 and 2021 at the global level.**

| | ASDR (/10^5) | | EAPC (95%UI) |
|---|---|---|---|
| | 1990 (95%CI) | 2021 (95%CI) | |
| Global | 33.013(24.316-40.657) | 22.641(17.000-27.762) | -1.369(-1.550--1.187) |
| Male | 36.672(26.086-45.876) | 27.045(19.661-33.580) | -1.103(-1.291--0.914) |
| Female | 29.849(22.504-36.783) | 18.878(14.132-23.825) | -1.666(-1.849--1.483) |
| SDI | | | |
| High SDI | 12.744(9.597-15.205) | 5.490(4.012-6.821) | -3.008(-3.148--2.868) |
| High-middle SDI | 36.835(27.089-45.359) | 21.720(15.637-27.406) | -2.083(-2.449--1.716) |
| Middle SDI | 48.668(35.089-61.945) | 32.212(23.783-40.099) | -1.366(-1.583--1.148) |
| Low-middle SDI | 39.670(28.943-49.446) | 29.953(22.732-36.527) | -0.942(-1.009--0.875) |
| Low SDI | 45.455(31.853-57.880) | 32.911(23.898-41.670) | -1.096(-1.162--1.031) |
| GBD region | | | |
| Andean Latin America | 12.244(8.139-16.883) | 7.261(4.913-9.873) | -1.760(-2.028--1.492) |
| Australasia | 8.609(6.467-10.294) | 3.488(2.485-4.445) | -3.047(-3.171--2.923) |
| Caribbean | 25.286(17.873-31.717) | 17.708(12.539-22.609) | -1.148(-1.250--1.047) |
| Central Asia | 43.837(32.968-52.893) | 30.333(23.104-36.611) | -1.544(-1.980--1.107) |
| Central Europe | 36.474(28.409-42.564) | 13.324(10.277-15.820) | -3.918(-4.191--3.645) |
| Central Latin America | 14.272(10.472-17.368) | 7.911(5.829-9.831) | -2.379(-2.534--2.223) |
| Central Sub-Saharan Africa | 60.243(40.551-83.342) | 45.102(31.977-63.356) | -1.078(-1.197--0.958) |
| East Asia | 66.917(47.426-88.460) | 38.477(27.238-50.643) | -1.860(-2.239--1.478) |
| Eastern Europe | 23.790(18.260-27.854) | 14.438(10.979-17.212) | -2.505(-3.028--1.980) |
| Eastern Sub-Saharan Africa | 54.722(38.121-70.289) | 40.373(29.020-50.846) | -1.106(-1.167--1.045) |
| High-income Asia Pacific | 19.321(14.414-23.407) | 5.418(3.945-6.797) | -4.362(-4.588--4.135) |
| High-income North America | 6.451(4.876-7.757) | 4.701(3.338-5.925) | -1.331(-1.591--1.069) |
| North Africa and Middle East | 31.032(21.813-39.616) | 14.608(10.512-18.458) | -2.623(-2.748--2.498) |
| Oceania | 56.732(36.467-79.003) | 49.377(32.340-68.228) | -0.458(-0.520--0.396) |
| South Asia | 30.069(21.245-39.026) | 22.298(16.056-28.405) | -0.992(-1.065--0.919) |
| Southeast Asia | 66.723(47.650-82.640) | 51.080(37.522-62.797) | -0.811(-0.969--0.652) |
| Southern Latin America | 20.569(14.376-25.863) | 8.804(6.581-10.555) | -2.541(-2.615--2.468) |
| Southern Sub-Saharan Africa | 30.555(21.833-38.031) | 32.516(24.791-39.569) | 0.330(-0.207-0.870) |
| Tropical Latin America | 27.358(19.999-33.140) | 9.862(7.263-12.017) | -3.403(-3.484--3.323) |
| Western Europe | 12.044(9.216-14.148) | 4.509(3.269-5.513) | -3.374(-3.487--3.261) |
| Western Sub-Saharan Africa | 47.361(32.879-62.519) | 35.550(25.489-44.849) | -0.877(-0.972--0.782) |

ASDR, Age-standardized death rate; HICH, Hypertensive intracerebral hemorrhage; EAPC, Estimated annual percent change; SDI, socio-demographic index; GBD, Global Burden of Disease.

(EAPC = −1.666; 95% CI:-1.849--1.483) (Table 1) and age-standardized DALYs rate (EAPC = −1.740; 95% CI:-1.909--1.570) (Table 2) was more pronounced in females than in males, which to some extent reflects that the burden of disease is heavier for males than for females.

### Different countries estimate the burden of death and DALYs from HICH

In 2021, the ASDR for HICH across countries ranged from 2.241 to 99.179 per 100,000 people. The top three countries with the highest ASDRs were observed in Montenegro (99.179), the Republic of Nauru (91.452), and the Republic of Mozambique (82.148), while the lowest were in the Swiss Confederation (2.880), the Republic of

**Table 2. DALYs of HICH between 1990 and 2021 at the global level.**

| | The age-standardized DALYs rate (/10^5) | | EAPC (95%UI) |
|---|---|---|---|
| | 1990 (95%UI) | 2021(95%UI) | |
| **Global** | 768.148(563.149-949.548) | 521.085(388.700-638.078) | -1.401(-1.568--1.234) |
| **Male** | 868.035(618.391-1087.285) | 631.451(459.317-781.567) | -1.137(-1.308--0.966) |
| **Female** | 676.626(504.428-836.053) | 419.803(317.287-532.151) | -1.740(-1.909--1.570) |
| **SDI** | | | |
| High SDI | 295.453(222.951-354.117) | 125.310(91.772-154.602) | -3.069(-3.206--2.932) |
| High-middle SDI | 809.827(594.195-1006.343) | 466.996(334.346-589.160) | -2.161(-2.523--1.797) |
| Middle SDI | 1061.230(760.750-1351.482) | 688.260(504.362-856.273) | -1.428(-1.621--1.234) |
| Low-middle SDI | 955.260(691.465-1193.487) | 717.947(540.475-875.761) | -0.949(-1.018--0.879) |
| Low SDI | 1092.491(766.995-1393.341) | 787.463(576.417-996.432) | -1.145(-1.215--1.074) |
| **GBD region** | | | |
| Andean Latin America | 295.331(190.333-412.490) | 170.667(115.352-234.332) | -1.868(-2.155--1.581) |
| Australasia | 178.602(134.252-213.903) | 67.129(49.547-84.627) | -3.322(-3.486--3.158) |
| Caribbean | 635.377(446.805-809.926) | 461.281(324.995-592.988) | -1.002(-1.124--0.881) |
| Central Asia | 1024.012(773.750-1234.462) | 669.692(511.302-812.975) | -1.799(-2.231--1.365) |
| Central Europe | 820.932(640.428-959.693) | 289.232(222.929-344.034) | -4.077(-4.371--3.782) |
| Central Latin America | 331.996(242.353-405.786) | 184.810(135.716-231.109) | -2.375(-2.549--2.201) |
| Central Sub-Saharan Africa | 1410.153(955.954-1980.667) | 1020.920(719.580-1404.335) | -1.197(-1.312--1.082) |
| East Asia | 1308.711(916.912-1747.359) | 752.505(529.899-990.554) | -1.857(-2.212--1.501) |
| Eastern Europe | 582.381(449.074-683.376) | 373.388(279.575-445.768) | -2.364(-2.940--1.784) |
| Eastern Sub-Saharan Africa | 1299.109(902.926-1699.402) | 953.007(691.425-1199.946) | -1.141(-1.209--1.074) |
| High-income Asia Pacific | 447.744(333.250-543.740) | 130.396(94.762-160.950) | -4.319(-4.552--4.085) |
| High-income North America | 149.631(111.993-180.947) | 105.383(73.717-132.210) | -1.420(-1.642--1.198) |
| North Africa and Middle East | 715.276(498.900-905.125) | 335.213(239.738-423.753) | -2.669(-2.775--2.563) |
| Oceania | 1293.504(836.503-1813.478) | 1168.148(770.577-1626.472) | -0.307(-0.394--0.221) |
| South Asia | 735.201(524.332-947.709) | 540.458(396.091-685.976) | -1.021(-1.090--0.953) |
| Southeast Asia | 1576.427(1130.100-1952.146) | 1213.487(883.873-1493.431) | -0.781(-0.917--0.644) |
| Southern Latin America | 500.568(344.373-639.128) | 208.406(152.428-252.754) | -2.697(-2.768--2.626) |
| Southern Sub-Saharan Africa | 772.963(560.624-955.523) | 773.603(588.249-935.271) | 0.152(-0.372-0.679) |
| Tropical Latin America | 723.591(528.599-881.083) | 253.365(186.320-308.801) | -3.561(-3.645--3.478) |
| Western Europe | 261.136(199.221-307.179) | 89.260(64.607-107.801) | -3.717(-3.839--3.594) |
| Western Sub-Saharan Africa | 1107.616(776.644-1453.038) | 833.257(606.188-1050.052) | -0.868(-0.979--0.756) |

DALYs, Disability-adjusted life years; HICH, Hypertensive intracerebral hemorrhage; EAPC, Estimated annual percent change; SDI, socio-demographic index; GBD, Global Burden of Disease.

Singapore (3.373), and Canada (3.723) (Fig 1A). Similarly, the age-standardized DALYs rate for HICH varied from 42.419 to 2476.318 per 100,000 people. The top three countries with the highest age-standardized DALY rates were found in the Republic of Nauru (2476.318), the Republic of Mozambique (2083.400), and the Republic of Vanuatu (1881.039), whereas the lowest were in the Swiss Confederation (42.419), Ireland (59.413), and Canada (61.874) (Fig 1B). Interestingly, most of the countries with a high burden of disease are distributed in hot tropical regions. There is some consistency in the top three countries with the highest and lowest distribution of the ASDR and the age-standardized DALYs rate.

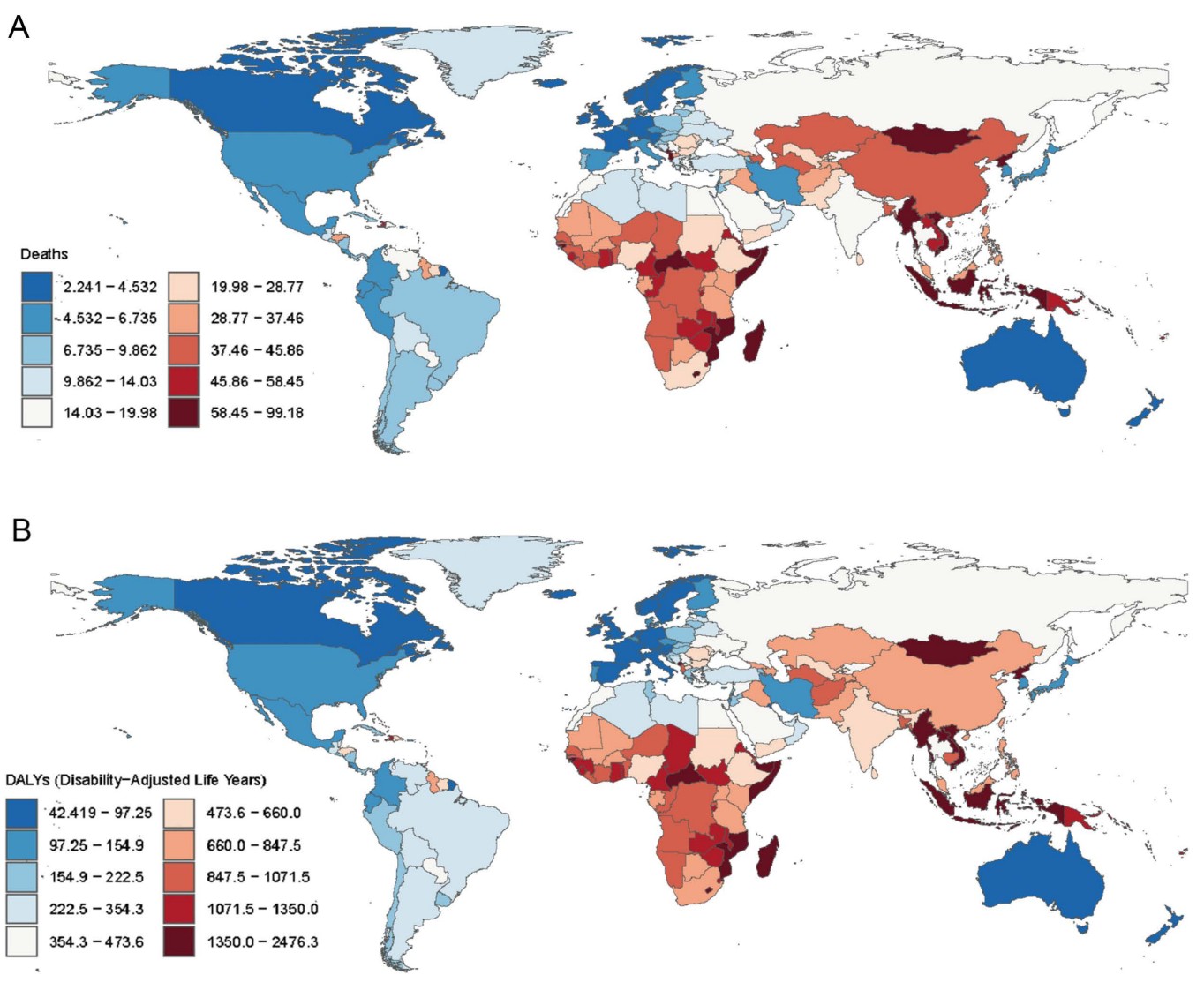

**Fig 1. Different countries' spatial distribution of HICH ASDR (A) and age-standardized DALYs rate (B) in 2021.**

### Different regions estimate the burden of death and DALYs from HICH

In 2021, high ASDRs and age-standardized DALY rates for HICH were observed in most of the regions in Central Africa, South Africa, Central Asia, East Asia, and Southeast Asia, most of which are coastal regions, whereas the rates were lower in these regions in the Americas, Europe, and Oceania. Moreover, the distribution of ASDRs and age-standardized DALYs rates for HICH were the same for the top three regions with the highest and lowest distributions. That is, Southeast Asia, Oceania, and Central Sub-Saharan Africa were the three regions with the highest ASDRs (Table 1) and age-standardized DALYs rates (Table 2) for HICH, while Australasia, Western Europe, and High-income North America were the three regions with the lowest ASDRs (Table 1) and age-standardized DALYs rates (Table 2).

From 1990 to 2021, the ASDR and the age-standardized DALYs rate for HICH declined in most regions. The most significant declines in ASDRs were observed in High-income Asia-Pacific (EAPC = −4.362, 95% CI: −4.588--4.135), Central Europe (EAPC = −3.918, 95% CI: −4.191--3.645), and tropical Latin America (EAPC = −3.403, 95% CI: −3.484–3.323)

(Table 1). Similarly, the largest declines in age-standardized DALYs rates were seen in High-income Asia-Pacific (EAPC = −4.319, 95% CI: −4.552--4.085), Central Europe (EAPC = −4.077, 95% CI: −4.371--3.782), and Western Europe (EAPC = −3.717, 95% CI: −3.839--3.594) (Table 2). Conversely, the ASDR (EAPC = 0.330, 95% CI: −0.207–0.870) (Table 1) and age-standardized DALYs rate (EAPC = 0.152, 95% CI: −0.372–0.679) (Table 2) in Southern Sub-Saharan Africa showed minimal overall change and even a trend of increase (Fig 2). This is consistent with the trend in the distribution of EAPC of ASDR and age-standardized DALYs rate for the 21 GBD districts combined with the five different levels of SDI districts for the period 1990–2021 (Fig 3).

## Different age and sex estimates of the burden of death and DALYs from HICH

In the 21 GBD regions, males exhibited higher ASDRs and age-standardized DALYs rates than females overall. However, in Western Sub-Saharan Africa, females had higher ASDRs and age-standardized DALYs rates compared to males (Fig 4). In 2021, global ASDR from HICH peaks in the 90–94 age group and is higher in males than in females in all age groups. Deaths peak in the 65–69 age group, increasing with age at <65 years and decreasing with age at >69 years (Fig 5A). Global age-standardized DALYs rates for HICH for both men and women peaked in the 85–89 age group and declined with age after age 89 years, and age-standardized DALYs rates were higher for men than for women in all age groups (Fig 5B). Additionally, the number of deaths and DALYs were higher for males than females up to the 80–84 age group, after which females had higher numbers in both metrics (Fig 5).

## SDI-related patterns

From 1990–2021, the ASDR and age-standardized DALYs rate for HICH showed an overall decreasing trend across the five different SDI regions (Fig 6). Notably, the ASDR and age-standardized DALYs rate in high-middle SDI and middle SDI areas exhibited an increasing trend in 1998, followed by a sharp decline after 2004 (Fig 6).

We also explored the relationship between ASDRs, age-standardized DALYs rates, and SDI. The results, shown in Fig 7 and 8, indicate that there is a significant negative correlation between SDI and ASDR ($\rho = -0.687, P < 0.05$) and age-standardized DALYs rate ($\rho = -0.700, P < 0.05$) at the geographic level (Fig 7), and that there is a significant negative correlation between SDI and ASDR ($\rho = -0.762, P < 0.05$), and age-standardized DALYs rate ($\rho = -0.769, P < 0.05$) at the country level (Fig 8), suggesting that geographic areas or countries with higher SDIs may have lower ASDRs and age-standardized DALYs rates for HICH. As shown in Fig 7, at the geographic level, from 1990 to 2021, East Asia, Southeast Asia, Australasia, Oceania, Central Asia, and High-income Asia Pacific have ASDRs, and age-standardized DALYs rates are all higher than expected. Fig 8 shows a negative correlation between SDI and ASDRs, age-standardized DALYs rates at the country level, with a number of countries having higher levels of ASDRs, age-standardized DALYs rates than would be expected based on the SDI from 1990 to 2021, such as Nauru, Mozambique, Vanuatu, and Indonesia.

## The average annual percentage change (AAPC) time trends and BAPC modeling

The AAPC time trend of ASDR (AAPC = −0.336, 95%CI: −0.352--0.319) and the age-standardized DALYs rate (AAPC = −7.953, 95%CI: −8.221--7.686) showed an overall decreasing trend. However, the AAPC time trend of ASDR exhibited an increasing trend between 1999 and 2003, followed by a significant decrease after 2003 (Fig 9A). In contrast, the AAPC time trend of the age-standardized DALYs rate showed a slow decrease between 1994 and 2004, with a significant decline thereafter (Fig 9B).

Additionally, the BAPC model associated with ASDR indicated that ASDR from HICH increased progressively with age (Fig 10A) and trended downward in more recent years (Fig 10B). There was a gradual increase in ASDR for those born before 1935 and a gradual decrease in those born after 1935 (Fig 10C), suggesting a decreasing trend in ASDR (|Net drift| > 0.01) (Fig 10J). The BAPC model for age-standardized DALY rate showed a similar pattern, with rate increasing with age. Using 1955 as the cut-off point, the age-standardized DALYs rate for those born before 1955 gradually increased, while it continued to decline for those born after 1955 (Fig 11).

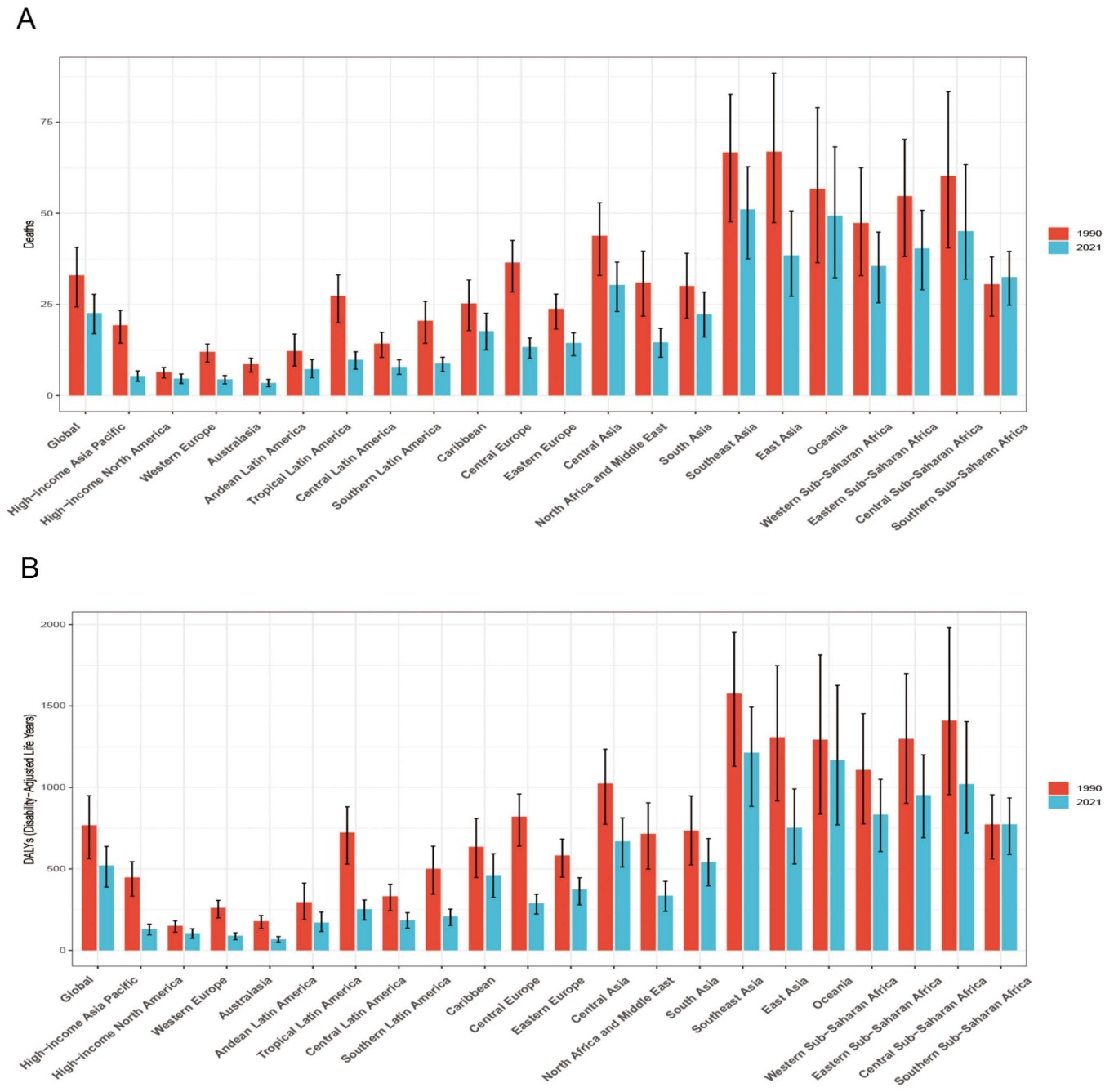

**Fig 2. Different regions' time distribution of HICH ASDR (A) and age-standardized DALYs rate (B).**

## Future projections of the global burden of death and DALYs from HICH

As shown in Fig 12, the ASDRs and age-standardized DALY rates for HICH across different age groups are projected to maintain an overall decreasing trend. Additionally, the BAPC model predictions indicate that the ASDR and age-standardized DALYs rate for HICH will continue to decline over the next 25 years (Fig 13). By 2050, the ASDR for HICH is

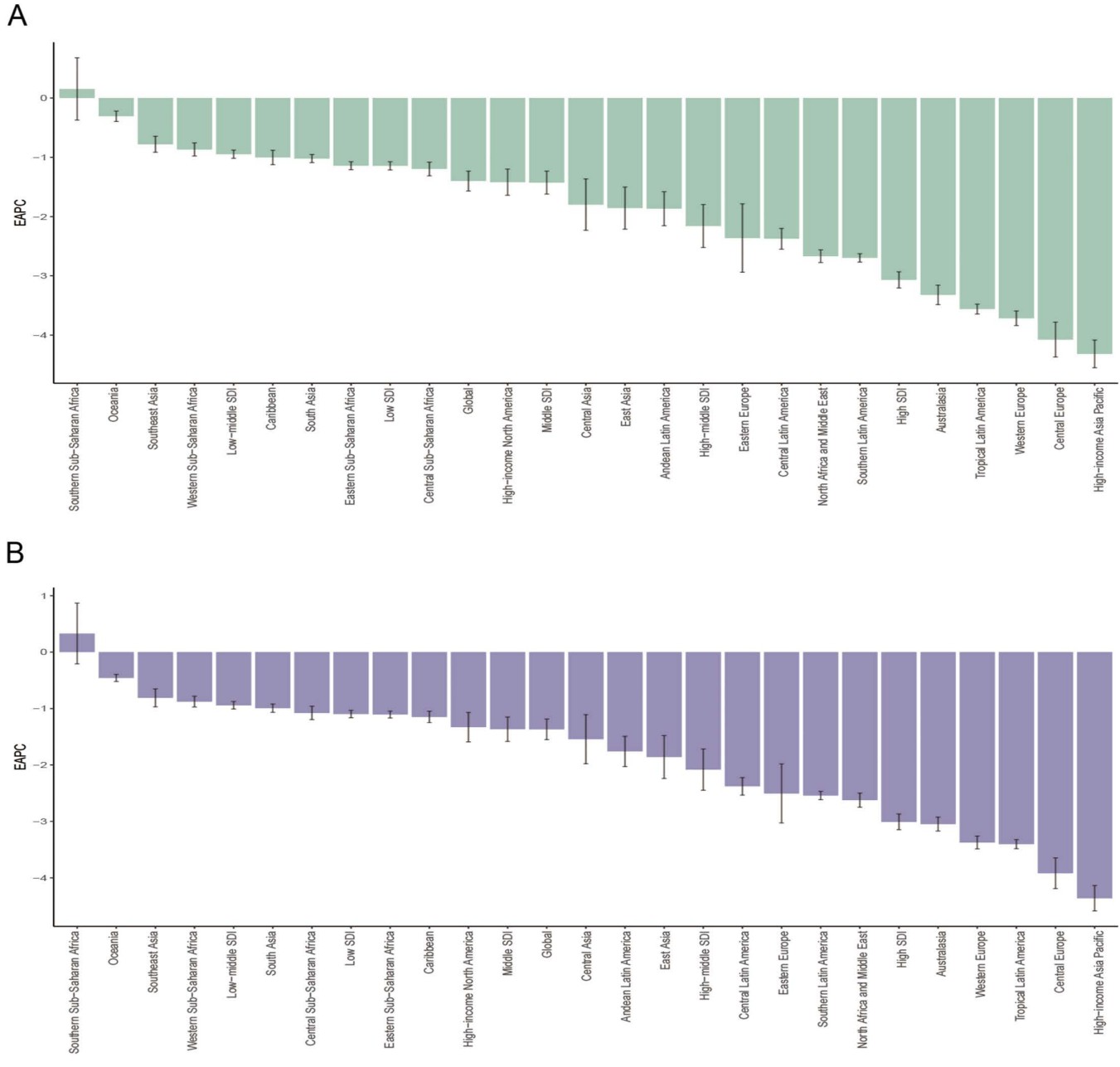

**Fig 3. 21 GBD regions and five different levels of SDI regions distribution of HICH ASDR (A) and age-standardized DALYs rate (B).**

projected to be 31.399 per 100,000 people, and the age-standardized DALYs rate is expected to be 758.805 per 100,000 people.

## Discussion

In the current study, we used ASDRs and age-standardized DALYs rates to measure the spatial and temporal trends in the burden of HICH over the period 1990–2021. We observed an overall declining trend in the global, national, and

A

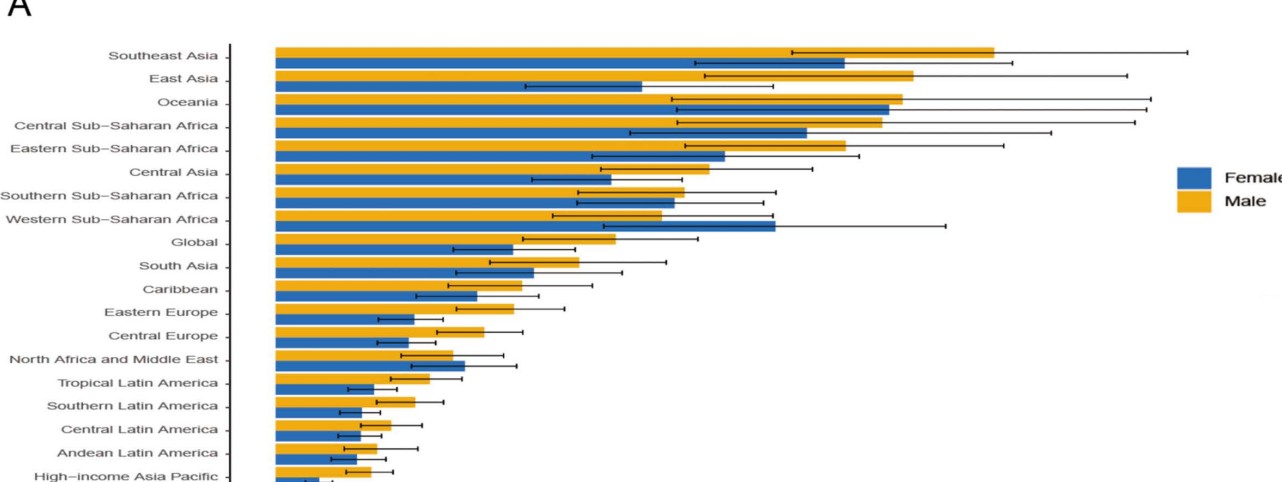

B

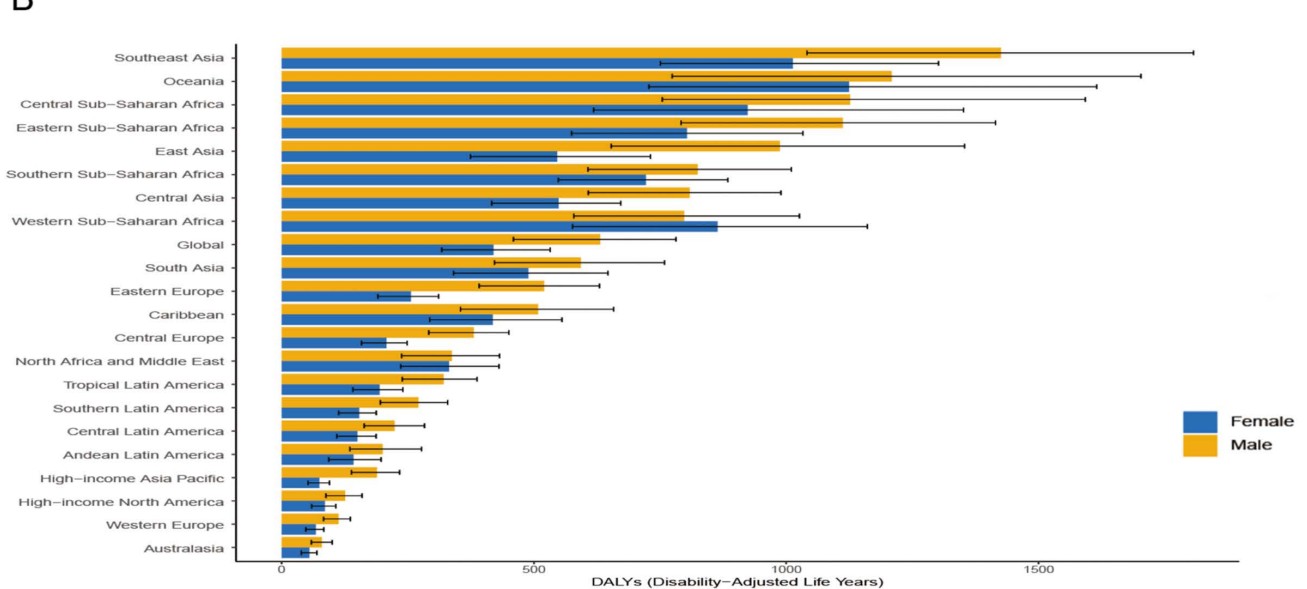

**Fig 4. Different sex regional distribution of HICH ASDR (A) and age-standardized DALYs rate (B).**

regional distribution of ASDRs and age-standardized DALYs rates for HICH since 1990, but with significant differences in the trends in spatial and temporal distribution. Trends in ASDRs and age-standardized DALYs rates for HICH are expected to decrease over the next three decades. The results of this study are generally consistent with previous studies showing an overall decreasing trend in the global burden of ICH [11,30]. This is the first study to explore HICH burden using updated 2021 GBD data stratified by SDI, age, and sex, and the findings have important value for the development of global health policy and the effective allocation of resources.

A

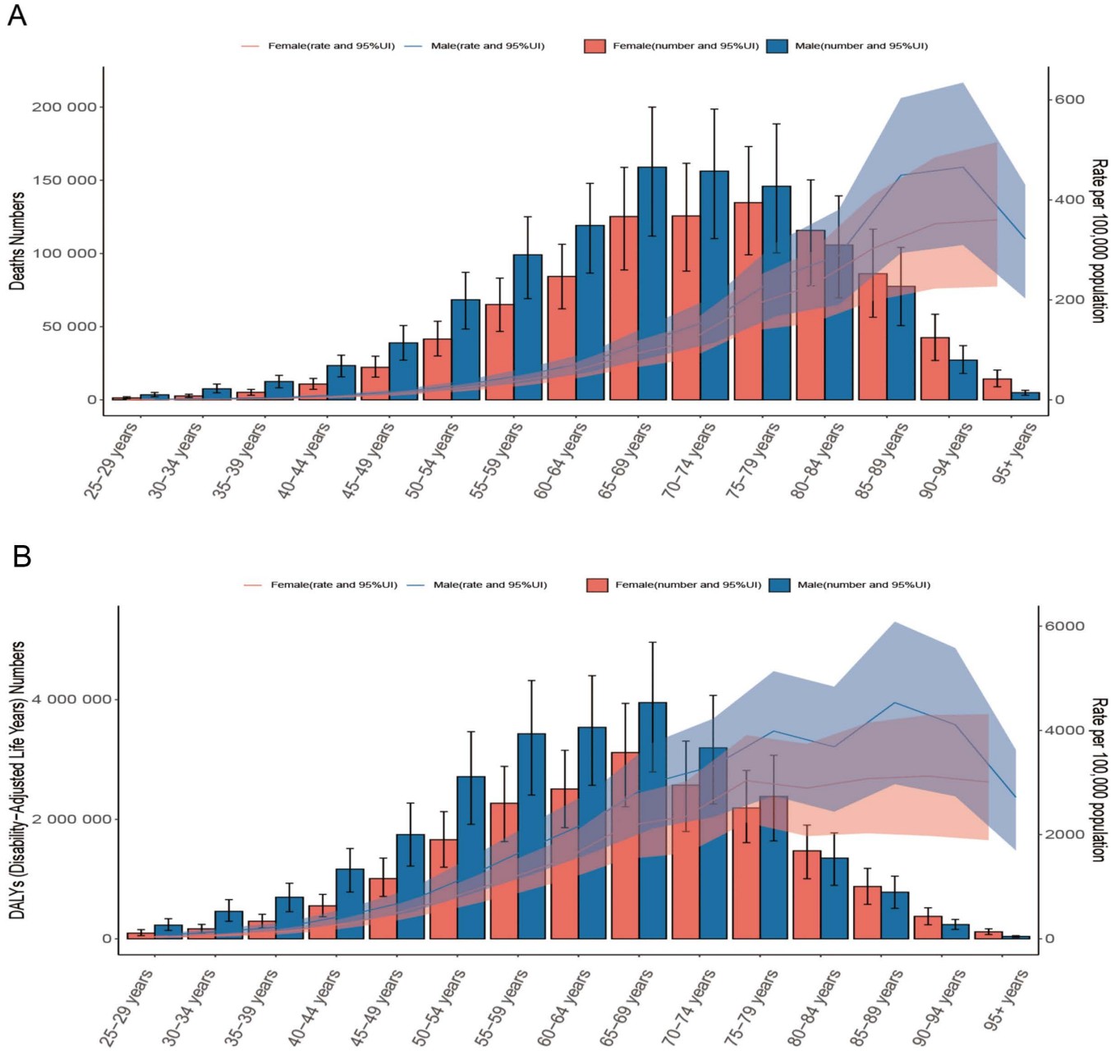

**Fig 5. Distribution of death (A) and age-standardized DALYs (B) with HICH in different age groups.**

Globally, the global ASDR and age-standardized DALYs rate for HICH showed a significant downward trend from 1990 to 2021, with an ASDR for HICH of 22.641 per 100,000 people in 2021, a decrease of 31.41792% from 1990, and an age-standardized DALYs rate of 521.085 per 100,000 people, a decrease of 32.16346% from 1990. This trend is particularly evident in High-income and upper-middle-income regions, thanks in large part to the well-developed medical and health facilities and better public health systems in those regions [31].In addition, the decreasing trends in ASDRs (EAPC = −1.666; 95% CI:-1.849--1.483) and age-standardized DALYs rates(EAPC = −1.740; 95% CI:-1.909--1.483) were

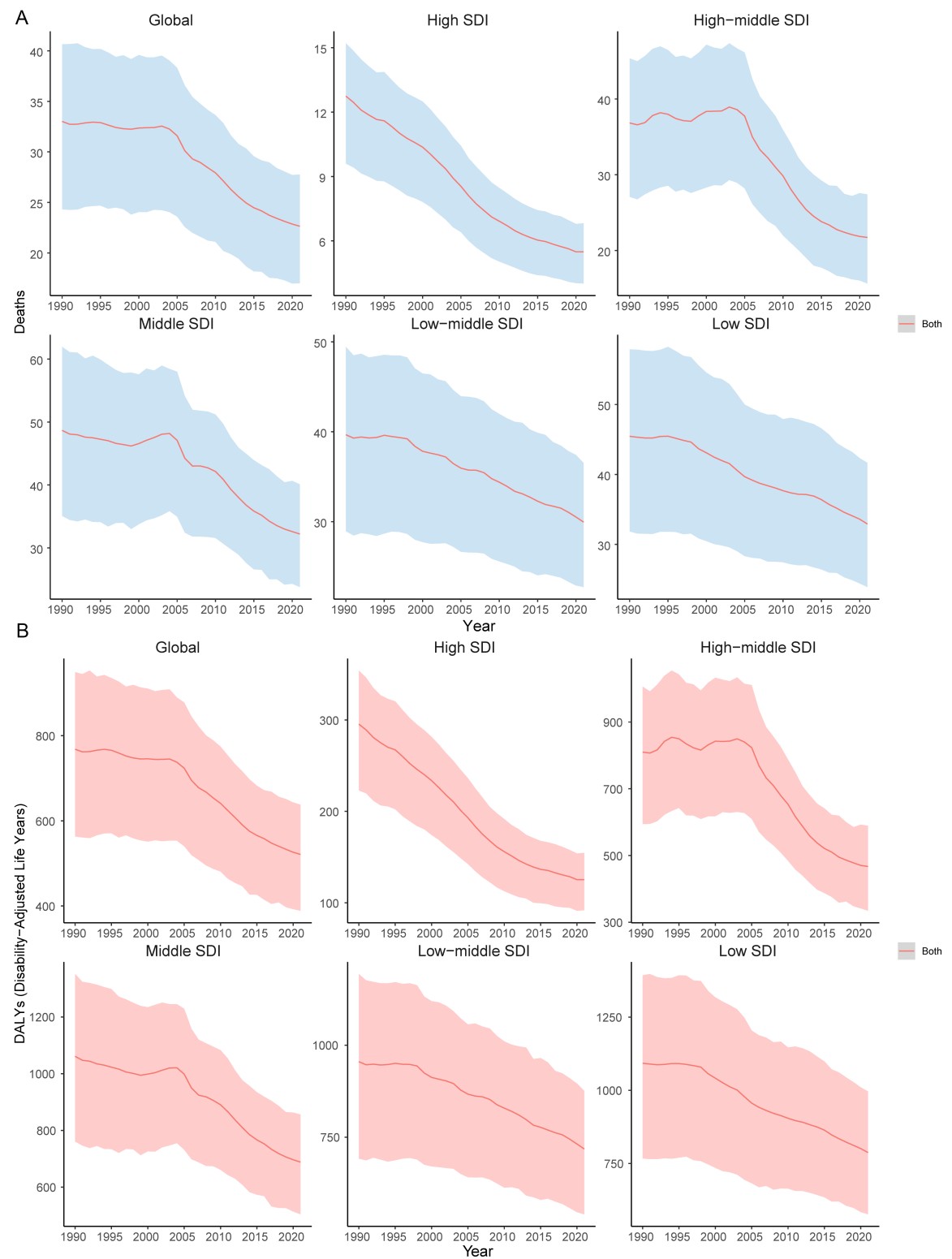

**Fig 6. Temporal trends of HICH in death (A) and age-standardized DALYs (B) at different SDI levels.**

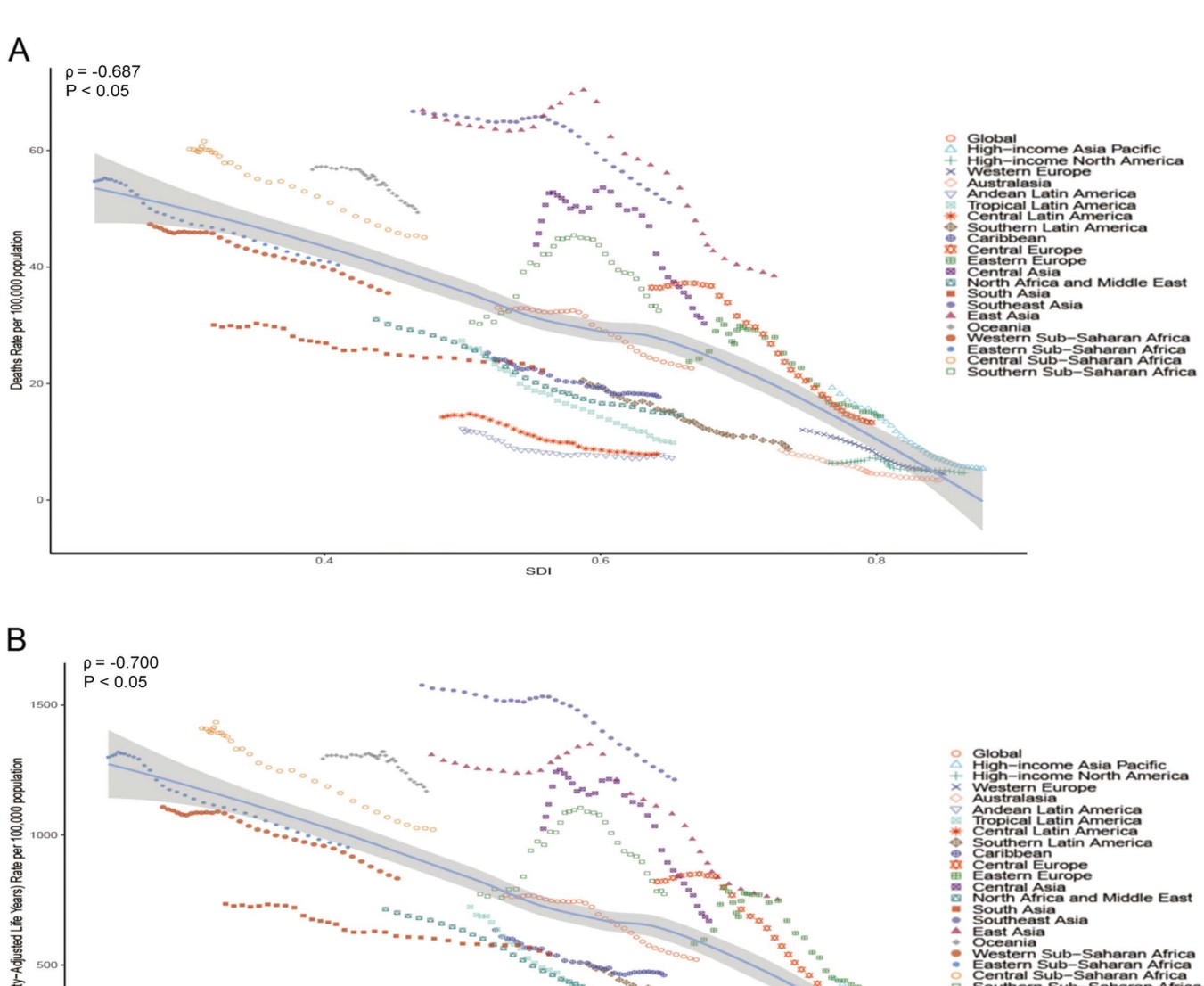

**Fig 7. Trends of HICH in ASDR (A) and age-standardized DALYs rate (B) in different regions at different levels of SDI.**

more pronounced in females than ASDRs(EAPC = −1.103;95% CI:-1.291--0.914 ) and age-standardized DALYs rates (EAPC = −1.137;95% CI:-1.308--0.966 ) in males, which may be due to gender-specific risk factors, with males experiencing spontaneous cerebral hemorrhage more frequently, at a younger age, and at a more severe site compared to females [32] Related studies have found that due to gonadal hormone levels, adenosine steroids in females, especially in young and middle-aged females, have a neuroprotective effect by decreasing early cerebral edema and neuroinflammation, thereby decreasing mortality rates [33,34]. Hormonal influences may explain some gender differences—estrogen's

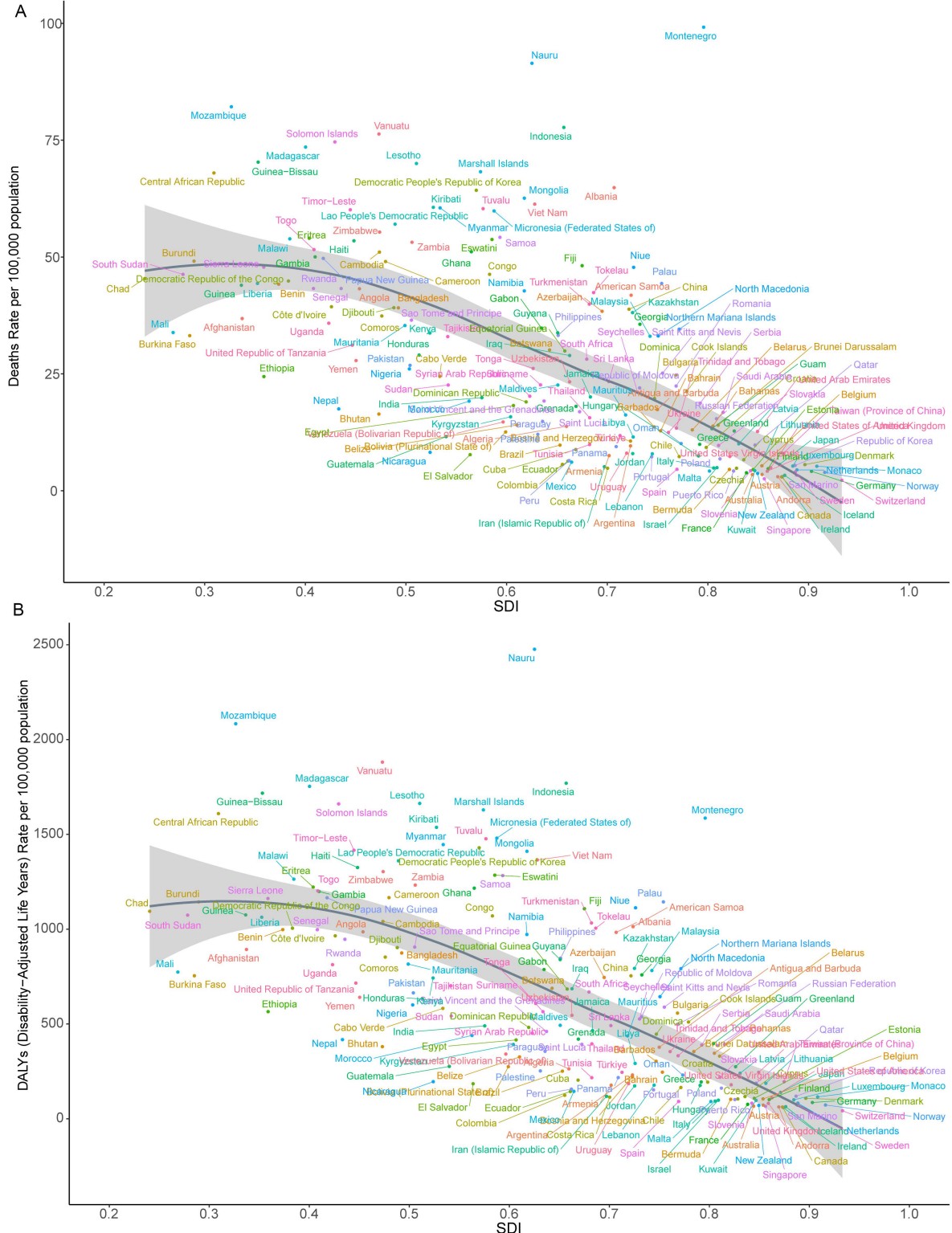

**Fig 8. Trends of HICH in ASDR (A) and age-standardized DALYs rate (B) in different countries at different levels of SDI.**

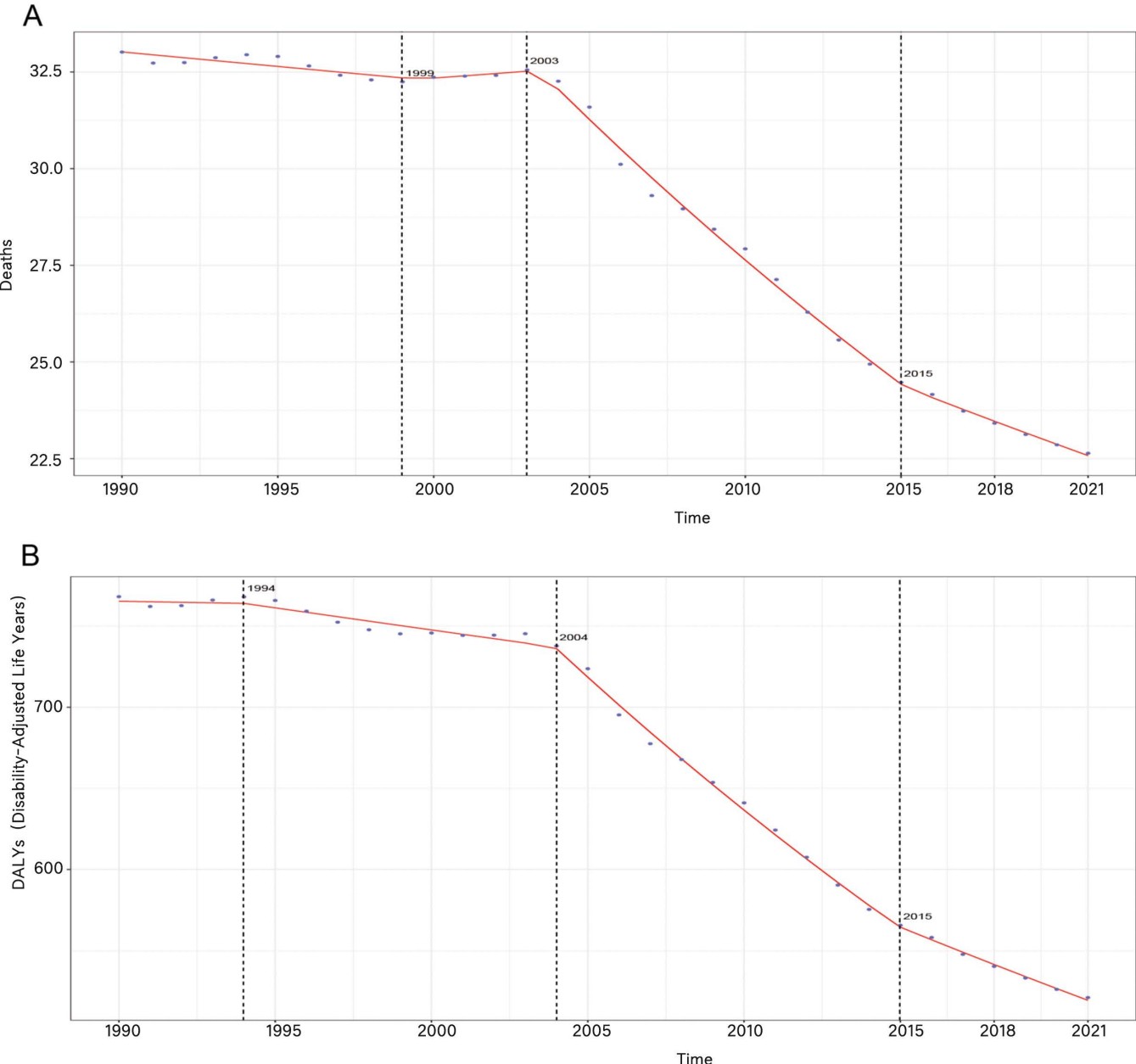

**Fig 9. The AAPC time trend of ASDR (A) and the age-standardized DALYs rate (B).**

neuroprotective effects in females delay ICH onset. In addition to biological factors, differences in gender trends are also reflected in access to health care and daily behaviors. Relevant studies have found that among hospital admissions, a higher proportion of men (85.6%) than women (74.7%) were admitted to acute stroke units or acute neurological intensive care units [35], mostly due to men's busy schedules, low health awareness, and failure to seek medical attention in a timely manner, which resulted in more serious conditions. Smoking and excessive alcohol consumption are important risk factors for stroke, with men having more current smoking (23.3% vs. 4.2%) and excessive alcohol consumption (16.5% vs. 1.3%) behaviors than women, which in turn increases the risk of stroke [36]. This highlights the need for

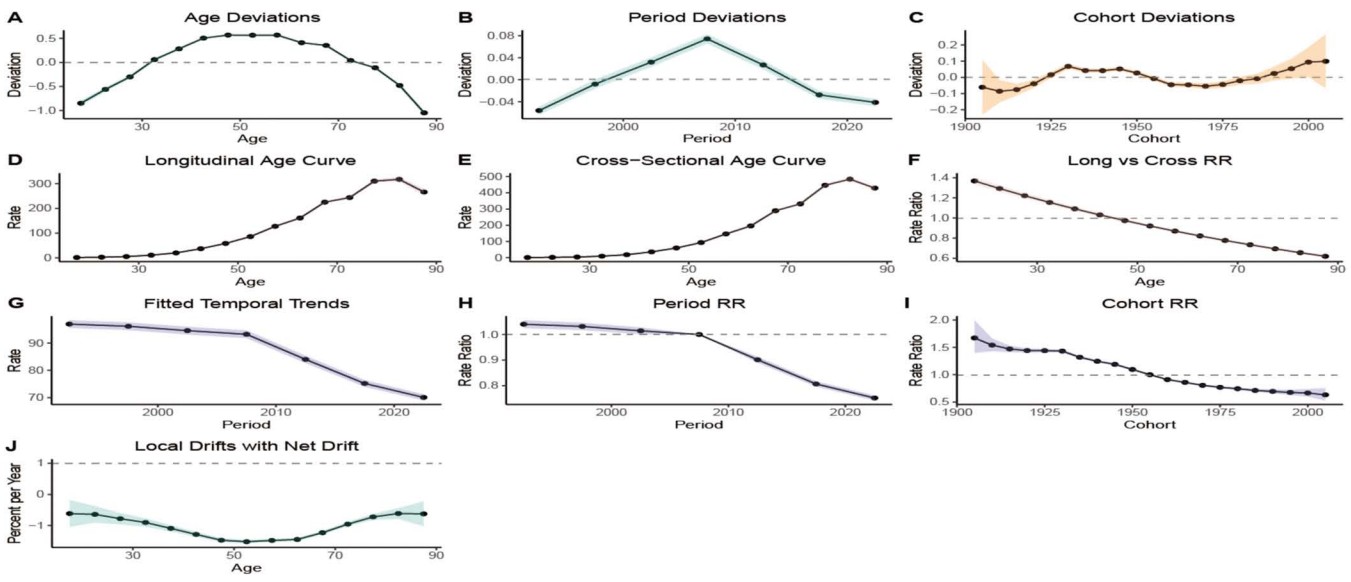

**Fig 10. The BAPC model of ASDR.**

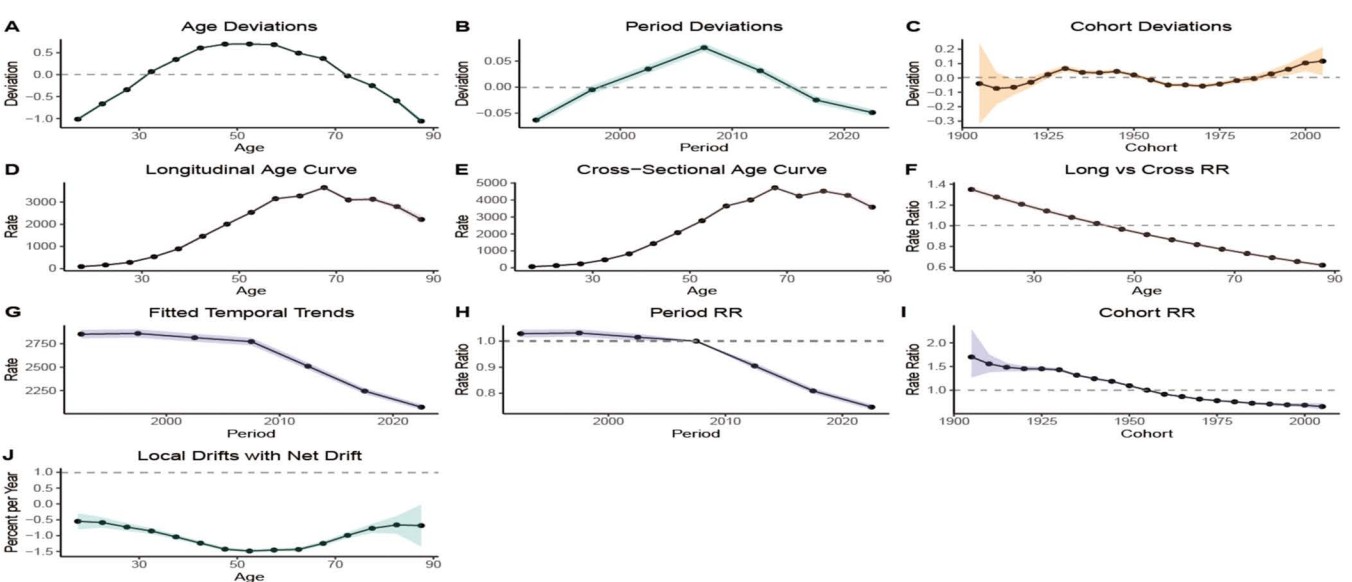

**Fig 11. The BAPC model of the age-standardized DALYs rate.**

gender-sensitive interventions, such as tailoring hypertension screening programs for men and women. Clinically, this may require incorporating gender-specific risk factor education into public health campaigns, especially for men with low health awareness and smoking and excessive drinking behaviors.

Shifting our focus to the national level in 2021, the Republic of Nauru's ASDR (91.452) and age-standardized DALYs rate (2476.318), the Republic of Mozambique's ASDR (82.148) and age-standardized DALYs rate (2083.400) were higher, while the Swiss Confederation had a lower ASDR (2.88) and age-standardized DALY rate (42.419), and Canada had a

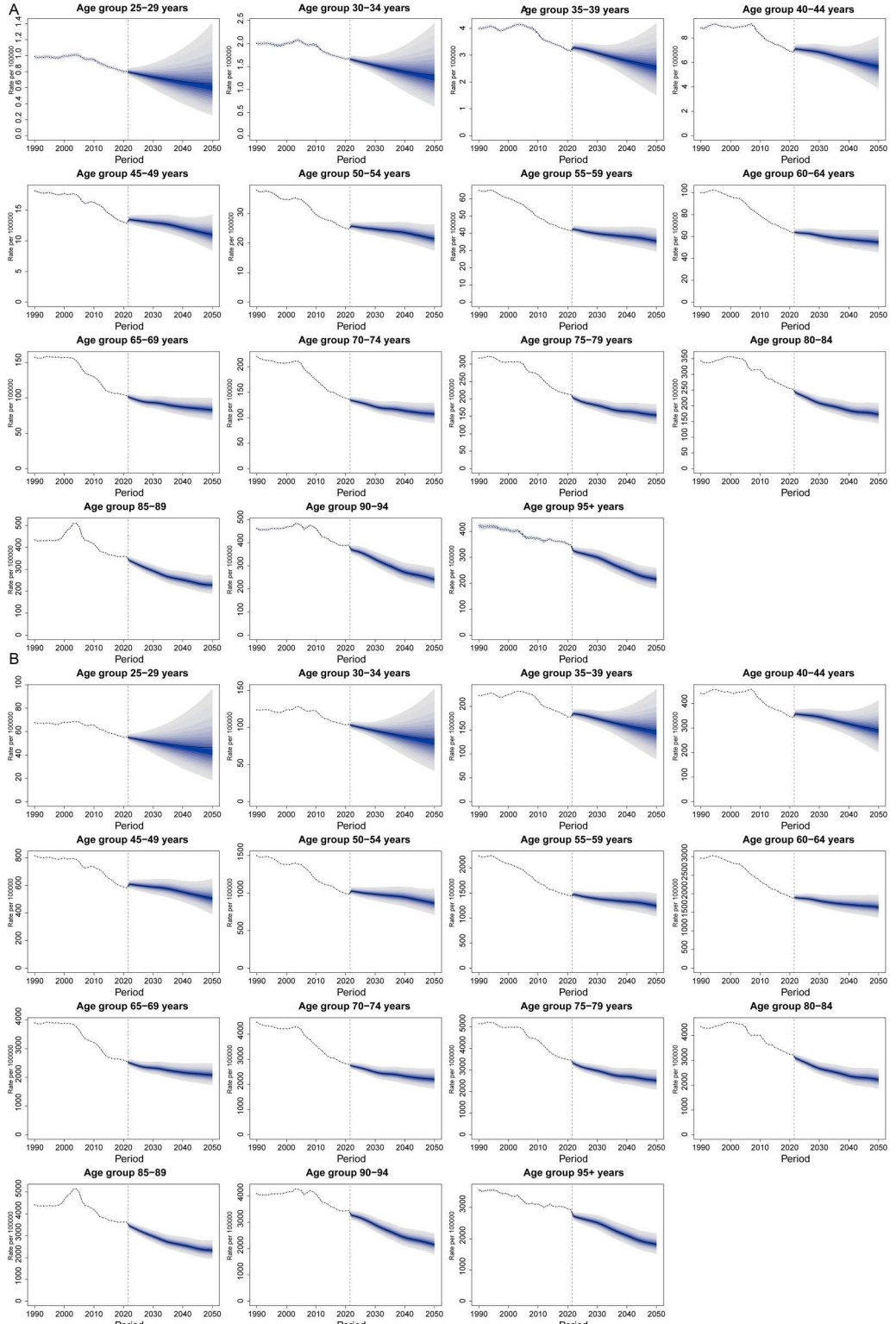

**Fig 12. Projected trends of HICH in ASDR (A) and age-standardized DALY rate (B) for different age groups for the next 25 years.**

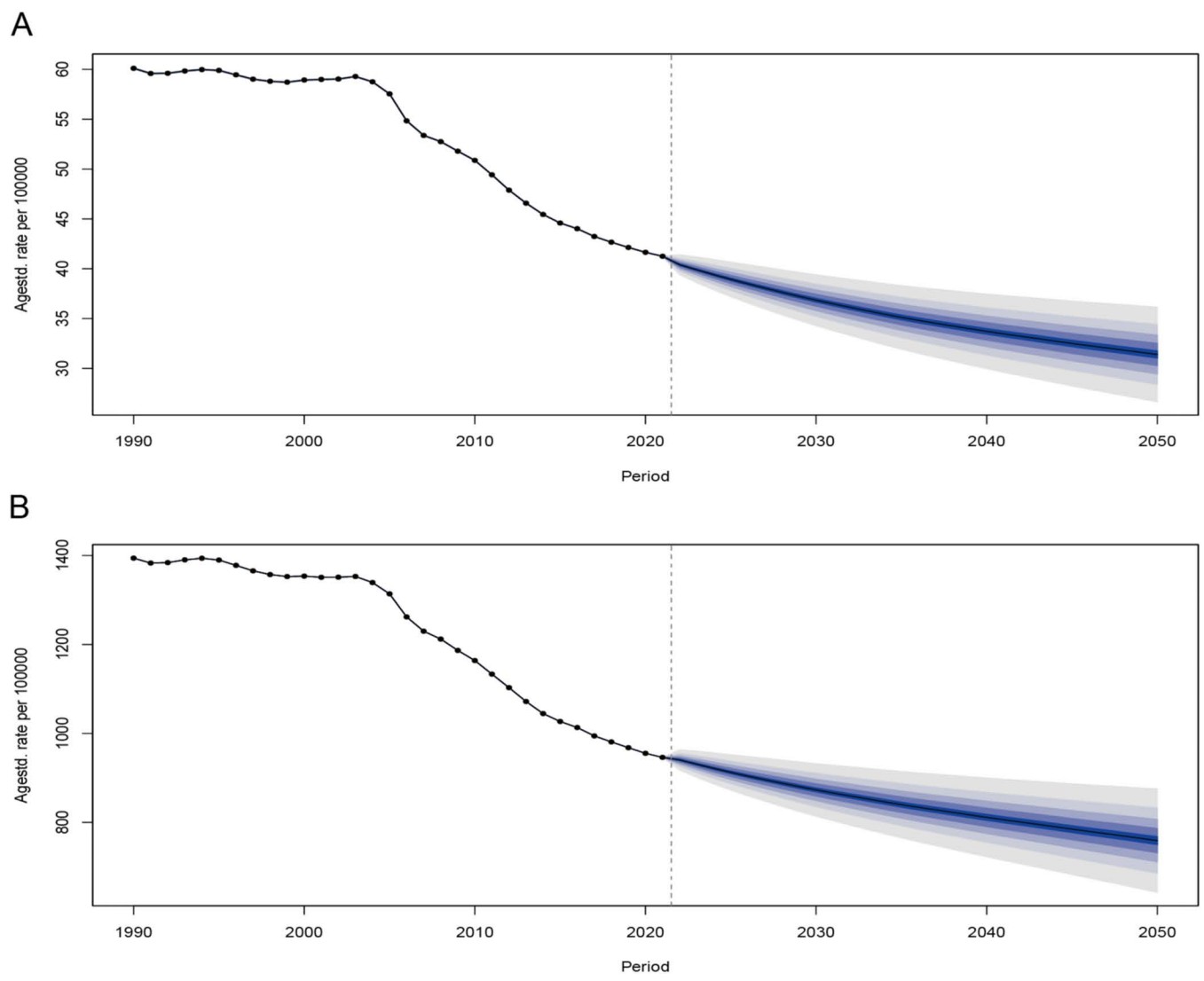

**Fig 13. ASDR (A) and age-standardized DALYs rate (B) BAPC projection model for the next 25 years.**

lower ASDR (3.723) and age-standardized DALY rate (61.874). These differences reflected both national differences in health services and disease prevention and the impact of economic and social development on the burden of disease [37].It is worth noting that the abnormally high values observed in Nauru and Mozambique may be due to limited data quality, small population size, prevalence of poor lifestyles [38], weak health care systems and disease surveillance and defense systems [39].

Examining regional perspectives, in 2021, Southeast Asia, Oceania, and Central Sub-Saharan Africa had the highest ASDRs for HICH at 51.080, 49.377, and 45.102 per 100,000 people, respectively, and these three regions also had the highest age-standardized DALYs rates for HICH at 1213.487, 1168.148, and 1020.920 per 100,000 people, respectively; Australasia, Western Europe, and High-income North America also had the lowest age-standardized DALYs for HICH at 3.488, 4.509, and 4.701 per 100,000 people, respectively, and these three regions also had the lowest age-standardized DALYs for HICH at 67.129, 89.260, and 105.383 per 100,000 people, respectively, reflecting regional differences in the

prevention and treatment of HICH. Regions with high ASDRs and age-standardized DALYs rates usually have a high prevalence of hypertension and its associated risk factors, insufficient health care resources, and low levels of literacy and education [40]. For example, Central, East, and Southeast Asia have a higher prevalence of hypertension, which is mostly associated with lifestyles such as a high-salt, high-fat diet, smoking, and lack of physical activity. Although the ASDR and age-standardized DALYs rate for HICH showed an overall decreasing trend from 1990 to 2021, Southern Sub-Saharan Africa showed little overall change or even an increasing trend. This may be due to the backward economic and social development of the region and the lack of medical facilities and resources. Future development should pay more attention to the changing trend of these underdeveloped regions and promote medical cooperation and exchanges between different income regions so as to better drive the enhancement of medical service and health care capacity in underdeveloped regions. In addition, global population aging has exacerbated the burden of ASDRs and age-standardized DALY rates in these regions to a certain extent. Strengthening medical infrastructure and public health services, such as integrating screening for HICH into existing hypertension programs, building regional telemedicine networks to serve rural areas, and training community health workers, are critical to reducing the burden of HICH in these regions.

When considering age and gender, in the 21 GBD regions, the ASDR and age-standardized DALYs rate for males are higher than those for females, but in Western Sub-Saharan Africa, the ASDR and age-standardized DALYs rate for females are higher than those for males. This is mostly due to the lack of healthcare infrastructure and resources in underdeveloped regions and limited cultural and educational levels, leading to untimely treatment or a poorer prognosis for women [41]. In 2021, the global ASDR for HICH reached its highest level in the 90–94 age group, and the age-standardized DALYs rate peaked in the 85–89 age group, with males having higher rates than females in all age groups. This is mostly due to the deep development of global aging and the influence of high-risk factors such as smoking, drinking, and obesity in most of the male patients [42]. An important cut-off point is the 80–84 age group, before which the number of deaths and DALYs is higher in males than in females, and 80–84 and beyond, when the number of deaths and DALYs is higher in females than in males. This is consistent with previous findings that men are approximately four years younger than women at the onset of ICH [42], mostly related to the neuroprotective effects of female gonadal hormones, which delay the onset of ICH in women by actively lowering blood lipid levels and directly altering the rapid vasorelaxation response of the blood vessel wall [43].

This study also explored the changes in ASDRs and age-standardized DALY rates in HICH at different SDI levels. The results of the study found that there was a significant negative correlation between SDI and ASDR and age-standardized DALYs at both the geographic and country levels ($\rho < 0$, $P < 0.05$), suggesting that geographic areas or countries with higher SDIs may have lower ASDR and age-standardized DALYs rate for HICH. This is mostly due to the fact that more economically and socially developed regions tend to have better medical facilities and higher health awareness, enabling more effective prevention and treatment of the disease, thus reducing the disease burden. This again emphasizes that the level of economic and social development has a significant impact on the disease burden. The results of the AAPC time trend and BAPC model indicate that, except for the temporary reversal of ASDR observed in 1999–2003, there has been an overall downward trend in ASDR, age-standardized DALYs rate for HICH since 1990, with a particularly significant decline during the period 2003–2015. Related studies have found that this is mostly due to improvements in care practices and medications [44].This temporary reversal stems largely from the overlap of multiple factors. Large-scale geopolitical conflicts (e.g., the 1999 Kosovo War [45], the 2003 Iraq War [46], etc.) severely disrupted local health care systems-destroying infrastructure, creating supply shortages, and displacing populations-thereby raising regional disease burdens and skewing global aggregates. At the same time, transitional health policy reforms in some countries (e.g., China's experimental restructuring of health insurance [47]) led to temporary fluctuations in the accessibility of services and the effectiveness of disease management during this period. These combined factors drove short-term upward deviations in the long-term downward trajectory.

Our projections for the future indicate that the ASDR for HICH in 2050 will be 31.399 cases per 100,000 people, and the age-standardized DALYs rate will be 758.805 cases per 100,000 people. Although the global ASDR and age-standardized DALYs rate for HICH are on a downward trend, significant disparities persist across countries, regions, SDI levels, and genders. This prediction underscores the urgent need to prioritize targeted interventions in low-SDI and high-burden regions to align with Sustainable Development Goal 3 (SDG 3), which aims to reduce premature mortality from non-communicable diseases by one-third by 2030. For instance, the projected burden highlights the necessity of integrating HICH prevention into global health initiatives such as the World Health Organization's Global Stroke Action Plan, particularly through initiatives like expanding hypertension screening in Central Sub-Saharan Africa and implementing gender-sensitive interventions to address male-female risk disparities. These findings also inform strategies for achieving universal health coverage (SDG 3.8), emphasizing the need to allocate resources toward healthcare infrastructure in regions with persistent high ASDRs, such as Southeast Asia and Oceania. By 2050, realizing these goals will require collaborative efforts to bridge care gaps, which our projections identify as critical for mitigating the ongoing global HICH burden.

## Strengths and limitations

The strength of this study lies in its systematic analysis of spatial and temporal trends in ASDRs and age-standardized DALYs rates for HICH by country, region, age, sex, and level of SDI using data from the GBD study from 1990 to 2021. We also projected the global burden of ASDR, age-standardized DALYs rates for HICH to 2050. To the best of our knowledge, this GBD-based study represents one of the more comprehensive efforts to date to analyze the global burden of HICH in terms of mortality rates, DALYs, and projected future trends.

However, as described in previous studies [48–51], there are some limitations to the present study: Firstly, there may be challenges in diagnosing and reporting hypertensive cerebral hemorrhagic disease, i.e., variations in diagnostic criteria, reporting standards, and data collection methods in different countries and regions can affect the accuracy of results. Second, when GBD collects and organizes regional data, there may be incomplete data for some regions. For example, some studies may not provide regional data, while others may provide data that are inconsistent with GBD's regional delineation. In addition, the GBD database incorporates a relatively wide range of data, and the quality and availability of data vary significantly across countries and regions, especially in low- and middle-income countries that mostly lack robust data registries. As a result, the regional burden of HICH may not be accurately estimated, thereby affecting the effectiveness of targeted interventions and resource allocation.

Third, GBD analyses rely on statistical models and assumptions to make inferences, which to some extent ignores real-world complexity factors and may not better reflect the lived experiences of people with HICH in different cultural contexts. In addition, the Bayesian approach to long-term forecasting is prone to deviation from the assumed conditions, and data quality and availability can be compromised by the difficulty of incorporating unpredictable events. Fourth, no risk factors for HICH were identified in the GBD database. HICH is a multifactorial disease, and its pathogenesis is related to genetics, environment, and lifestyle in addition to hypertension as a major factor. This study may not provide a comprehensive understanding of the risk factors for HICH, which to some extent hinders the development of effective prevention and control strategies. Finally, the impact of time should be considered. This analysis spanned a period of 30 years, and projections of the future burden of HICH were mostly based on current trends and patterns, during which major developments in medical technology, treatment protocols, and public health policies may have had an impact on the burden of HICH, which to a certain extent ignored the possible intervening factors that may have been relevant in the passage of time. Research into the future should focus on assessing the impact of targeted health policies and equitable resource allocation while leveraging multidisciplinary approaches and advanced technologies—such as integrating electronic health records for real-time disease surveillance, applying AI in risk prediction models, developing AI-assisted diagnostic tools, and personalizing treatment recommendations through AI algorithms—to enhance HICH prevention, optimize clinical treatment options, and improve global health outcomes.

## Future directions

Recommendations for future research and policy are essential to maintain and enhance HICH disease control. First, further research should focus on understanding the underlying causes of ASDR and age – standardized DALYs rate anomalies in regions like sub-Saharan Africa, examining socio-cultural, economic, and environmental factors, such as how local cultures, religious beliefs, and economic development influence HICH morbidity and mortality. Second, exploring the interactive effects of genetics, diet, and pollution on HICH risk across different SDI levels can provide a basis for formulating targeted prevention strategies. Third, evaluating the impact of emerging technologies, such as telemedicine and AI-based diagnostic tools, on HICH management in resource-constrained areas is crucial for optimizing care delivery.

Policy actions must build on these research directions to enhance HICH control. Low- and middle-SDI regions need to strengthen healthcare infrastructure, which includes integrating HICH screening into existing hypertension programs, training community health workers, and establishing telemedicine networks to serve rural areas. Policy initiatives should also prioritize formulating national HICH prevention plans and integrating them into public health strategies, expanding health insurance coverage for low-income groups and increasing the reimbursement for HICH-related medical expenses, promoting international cooperation to share prevention technologies and best practices, and implementing socio-economic development programs such as air pollution control, the provision of safe drinking water, and the construction of public fitness facilities to address upstream risk factors.

## Conclusion

In summary, this study provides a comprehensive analysis of the current status and trends of the global burden of HICH using data from the 2021 GBD database. Despite the overall downward trend in ASDRs and age-standardized DALY rates globally, significant differences persist between regions and countries. It is essential to implement various measures to control HICH, particularly in less developed areas. These measures include increasing the screening, diagnosis, and treatment of the disease; enhancing investment in medical facilities and resources; and expanding the coverage of medical services. Additionally, international cooperation should be continuously strengthened to integrate resources from different regions and improve data collection and quality in low-resource settings, thereby contributing to the enhancement of global health.

## Author contributions

**Conceptualization:** Chao Zhang, Jiao Chen.

**Data curation:** Chao Zhang, Jiao Chen.

**Formal analysis:** Chao Zhang, Jiao Chen.

**Funding acquisition:** Dayong Ma, Jing Li.

**Investigation:** Chao Zhang, Jiao Chen.

**Methodology:** Chao Zhang, Jiao Chen.

**Project administration:** Chao Zhang, Jiao Chen.

**Software:** Chao Zhang, Jiao Chen.

**Supervision:** Chao Zhang, Jiao Chen, Linjing Song, Juwei Dong, Qianqian Hu.

**Validation:** Chao Zhang, Jiao Chen.

**Visualization:** Chao Zhang, Jiao Chen.

**Writing – original draft:** Chao Zhang, Jiao Chen.

**Writing – review & editing:** Chao Zhang, Jiao Chen, Linjing Song, Juwei Dong, Qianqian Hu, Dayong Ma, Jing Li.

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
