## [Decision Letter · Decision Letter 0]

19 May 2025

Dear Dr. Ma,

Thank you for submitting your manuscript to PLOS ONE. After careful consideration, we feel that it has merit but does not fully meet PLOS ONE’s publication criteria as it currently stands. Therefore, we invite you to submit a revised version of the manuscript that addresses the points raised during the review process.

**ACADEMIC EDITOR: The topic is of interest. However, the manuscript needs revision before further consideration.**

We look forward to receiving your revised manuscript.

Kind regards,

Thien Tan Tri Tai Truyen, M.D.

Academic Editor

PLOS ONE

Journal Requirements:

3. Please ensure that you refer to Figure 9 to 13 in your text as, if accepted, production will need this reference to link the reader to the figure.

4. Please upload a copy of Supporting Information Figure/Table/etc. “Supporting information” which you refer to in your text on page 27.

Reviewers' comments:

Reviewer's Responses to Questions

**Comments to the Author**

1. Is the manuscript technically sound, and do the data support the conclusions?

Reviewer #1: Yes

Reviewer #2: Yes

2. Has the statistical analysis been performed appropriately and rigorously?

Reviewer #1: Yes

Reviewer #2: I Don't Know

3. Have the authors made all data underlying the findings in their manuscript fully available?

Reviewer #1: Yes

Reviewer #2: Yes

4. Is the manuscript presented in an intelligible fashion and written in standard English?

Reviewer #1: Yes

Reviewer #2: Yes

Reviewer #1: The manuscript analyzes the global, regional, and national burden of hypertensive intracerebral hemorrhage (HICH) from 1990 to 2021 using data from the Global Burden of Disease (GBD) 2021 study. It focuses on age-standardized death rates (ASDR) and disability-adjusted life years (DALYs), examining trends across countries, regions, age groups, sexes, and sociodemographic index (SDI) levels. The study reports a significant decline in global ASDR (31.418%) and DALYs (32.163%) from 1990 to 2021, with projections to 2050 using a Bayesian age-period-cohort (BAPC) model indicating continued declines. The authors highlight persistent disparities in less developed regions and call for targeted prevention and treatment strategies to address these gaps.

Areas of improvement:

1. Limited Discussion of HICH-Specific Risk Factors:

While the manuscript identifies hypertension as the primary risk factor for HICH, it does not sufficiently explore other contributing factors (e.g., smoking, alcohol use, obesity) or their interactions with hypertension. The discussion of risk factors is brief and relies heavily on general statements.

Recommendation: Expand the discussion to include a detailed analysis of HICH-specific risk factors, supported by GBD data on attributable fractions or relevant literature. Discuss how these factors vary by region, sex, or SDI.

2. Inadequate Exploration of Data Limitations:

The manuscript acknowledges variability in data quality, particularly in low- and middle-income countries, but does not thoroughly discuss the implications of these limitations. For example, it does not address potential biases in HICH diagnosis, underreporting, or differences in healthcare access that may affect GBD estimates.

Recommendation: Dedicate a paragraph in the discussion to explore data limitations, including diagnostic accuracy, data completeness, and the ecological study design’s inability to infer causality. Discuss how these limitations may impact the findings and projections.

3. Overreliance on Descriptive Statistics:

The results section is heavily descriptive, presenting extensive numerical data (e.g., ASDR, DALYs, EAPC) without sufficient synthesis or interpretation. This makes the section dense and challenging to follow for readers unfamiliar with GBD studies.

Recommendation: Streamline the results section by focusing on key trends and using tables/figures to summarize data. Provide more interpretive commentary to highlight the significance of findings (e.g., why certain regions have higher burdens).

4. Insufficient Detail on BAPC Model:

The manuscript briefly describes the BAPC model but does not explain its assumptions, limitations, or how it handles uncertainty in projections. Readers unfamiliar with this method may struggle to assess its validity.

Recommendation: Expand the methods section to include a detailed explanation of the BAPC model, including its mathematical framework, assumptions (e.g., stationarity of trends), and how it accounts for future changes in risk factors or healthcare access. Discuss potential limitations of long-term projections.

5. Generic Policy Recommendations:

The policy recommendations (e.g., increasing screening, improving medical facilities) are broad and lack specificity regarding implementation, cost-effectiveness, or feasibility in low-resource settings. The manuscript does not address potential barriers, such as political or cultural resistance.

Recommendation: Provide more concrete policy recommendations, such as integrating HICH screening into existing hypertension programs, leveraging telemedicine for rural areas, or adopting cost-effective interventions like community health worker training. Discuss implementation challenges and potential solutions.

6. Inconsistent Temporal References:

The manuscript inconsistently refers to the study period (e.g., “2019” in Tables 1 and 2 instead of 2021, “32 years” vs. “1990–2021”). This creates confusion about the data’s temporal scope.

Recommendation: Standardize all temporal references to 1990–2021 and correct errors in tables (e.g., replace “2019” with “2021”). Ensure consistency in describing the study period throughout the manuscript.

7. Underutilized Supplementary Materials:

The manuscript references 13 supplementary figures but does not adequately integrate them into the main text. For example, S9–S13 figures are cited but not summarized, limiting their utility for readers.

Recommendation: Provide brief summaries of key supplementary findings in the main text or include a dedicated section describing the supplementary materials. Ensure all figures are clearly labeled and cross-referenced.

8. Minor Editorial Issues:

The manuscript contains minor typographical errors (e.g., “supplemalestary” in some sections, inconsistent spacing in references) and awkward phrasing (e.g., “more severe sites” for HICH in males). These detract from the manuscript’s professionalism.

Recommendation: Conduct a thorough proofreading to correct typos, standardize terminology, and improve sentence clarity. Ensure references adhere to PLOS ONE’s formatting guidelines.

Reviewer #2: I have reviewed the manuscript analyzing trends in hemorrhagic intracerebral hemorrhage (HICH) using Global Burden of Disease data. I believe that this work provides valuable epidemiological insights, but I think that several minor revisions are needed before publication.

Introduction:

The progression from global ICH burden to HICH specifics and then to GBD data trends is somewhat abrupt. I would fix this flow, although it is still clear in a way, but it would be further enhanced with more logical transitions.

I think the introduction is good, but in my opinion, it lacks a clear statement of the knowledge gap or why analysing HICH trends specifically by geography, SDI, age, and gender is important. This would enhance the introduction and emphasise the importance and strength of the study. I would highly suggest that you fix this.

My last suggestion for this section is the referencing of literature. Some references (e.g. [5], [6]) are cited vaguely, and it is unclear what specific data or findings they refer to. If you further emphasise using numbers or statements from the studies you have referenced, you can make the reader more intrigued to read your research, since you have built a strong introduction.

Methods:

Generally, this section is well written and explained. There are some areas I would like to draw attention to, and I am saying this because I am a first-time reader, so my comments might be due to my confusion rather than my knowledge.

The phrase "queried raw data" is vague and needs clarification. I asked myself, "Did the authors extract incidence, mortality, DALYs, or other variables from GBD 2021? Was it done through the GHDx tool or GBD Results Tool?" I would suggest elaborating on this section further to enhance the strength of the methods used to conduct this strong study.

While the SDI breakdown is correct, I am confused about how the SDI is assigned to each country. The method used to assign countries to each SDI category should be cited or briefly explained (e.g. based on the GBD-defined quintiles for each year). This would be enhanced if you added a statement with a reference stating that each SDI is assigned to the countries using WHO categorisations. Explaining or emphasising standards here is very important because of the study aim; otherwise, it will be very vague and subject to critiques.

Also, the BAPC model is mentioned, but I am confused about how this model was used, designed, or validated. This section is very concise and lacks clarity in some places. Again, I believe that emphasising this section by detailing the methodological approach is essential.

Instead of "a p-value < for 0.05…", I would say "A p-value < 0.05 was considered statistically significant." How was the EAPC calculated? This is also not clear.

Results:

The results state that trends are "significant”, but there is no mention of p-values or interpretation of CIs (although the CIs do not cross 0, indicating statistical significance). I would suggest adding the P value here.

Discussion and Conclusion:

"The results of this study are generally consistent with those of previous studies showing an overall decreasing trend in the global burden of cerebral haemorrhage; however, the authors have not provided the actual numbers, which weakens the comparison and decreases the study strength in adding value to the current literature. Other sentences with the same structure in this section also lack numbers or comparison details.

I also noticed The discussion is mostly descriptive and lacks in-depth causal interpretation.

I noticed that The limitations and strengths section is missing. You may have merged it with the text, but I would highly recommend that you do a separate section.

I believe this manuscript addresses an important topic with significant public health implications. A comprehensive approach examining variations by geography, SDI, age, and sex offers valuable perspectives for targeted interventions. With the suggested revisions to enhance methodological clarity, narrative flow, and interpretive depth, I believe this work will make a significant contribution to understanding global HICH patterns and trends.

I recommend this manuscript for publication following moderate revisions.

**Do you want your identity to be public for this peer review?** For information about this choice, including consent withdrawal, please see our Privacy Policy

Reviewer #1: No

Reviewer #2: **Yes: ** Areej Almutairi

---

## [Author Response · Author response to Decision Letter 1]

15 Jun 2025

Response to Reviewers

Manuscript ID number:

PONE-D-25-19666

Title of paper:

Global, regional, and national burden of hypertensive intracerebral hemorrhage, 1990 to 2021 and projections to 2050: Results from the Global Burden of Disease Study 2021

Dear Editor and Reviewers,

We are very grateful for the opportunity to revise the manuscript entitled “Global, regional, and national burden of hypertensive cerebral hemorrhage from 1990 to 2021 and projections to 2050: results from the Global Burden of Disease Study 2021” and thank the reviewers for their insightful comments. These comments were invaluable and very helpful in revising and improving the paper, as well as providing important guidance for our research. In the submitted manuscript “Revised Manuscript with Track Changes”, we have kept all the traces of changes and made detailed corrections to the comments of the journal, reviewer 1, and reviewer 2, which are prominently highlighted in blue, yellow, and green, in that order. Below, we have provided a detailed response to each of the reviewers' comments and listed the number of lines of changes (for “Revised Manuscript with Track Changes” manuscripts) for easy reference. In addition, we have completely revised the entire manuscript. In this response letter, the reviewers' comments are italicized, and our corresponding changes and additions to the manuscript are highlighted in red. We have made every effort to make all revisions clear, and we hope that the revised manuscript will meet the requirements for publication.

In addition, there is a question for the editors to note that "the study was supported by the National Natural Science Foundation of China (82274458)" has been added to the manuscript funding section, and this project has also been added to the system. We apologize for any inconvenience caused, and please contact us if there is anything unreasonable.

Journal Requirements:

We have checked and revised the style requirements and file naming requirements of the manuscript according to the PLOS ONE style template provided by you to ensure that the revised manuscript meets the requirements of your journal.

We have included a full ethical statement in the “Methods” section of the manuscript document. The full name of the ethics committee that approved or exempted the study has also been indicated and the exemption of the study from obtaining informed consent has been explained. In the “Revised Manuscript with Track Changes”, lines 121-126.

3. Please ensure that you refer to Figure 9 to 13 in your text as, if accepted, production will need this reference to link the reader to the figure.

Figures 9 to 13, referred to in the text, have been redeployed and elaborated in order to better connect the reader to the figures. In the “Revised Manuscript with Track Changes”, lines 339-360.

4. Please upload a copy of Supporting Information Figure/Table/etc. “Supporting information” which you refer to in your text on page 27.

We are going to elaborate on this issue, due to our misinterpretation, the two tables with 13 images covered in the main text were written into the supplementary material, i.e., the “Supporting Information” on page 27 in the first version of the submitted manuscript is the same as the charts and graphs in the main text and no new charts/tables, etc., of supporting information have been added, so the supplementary material This section is not available to us and this section should be deleted. We apologize for any confusion caused to you. In the “Revised Manuscript with Track Changes”, lines 809-832.

Response to reviewer #1:

Dear Reviewer #1,

Thank you for your constructive comments on my manuscript. I appreciate your time and effort in reviewing my paper and providing valuable suggestions. I have revised manuscript according to your feedback and addressed each point in detail below.

Thank you again for reviewing the manuscript, and I hope my revisions meet your requirements.

Reviewer #1: The manuscript analyzes the global, regional, and national burden of hypertensive intracerebral hemorrhage (HICH) from 1990 to 2021 using data from the Global Burden of Disease (GBD) 2021 study. It focuses on age-standardized death rates (ASDR) and disability-adjusted life years (DALYs), examining trends across countries, regions, age groups, sexes, and sociodemographic index (SDI) levels. The study reports a significant decline in global ASDR (31.418%) and DALYs (32.163%) from 1990 to 2021, with projections to 2050 using a Bayesian age-period-cohort (BAPC) model indicating continued declines. The authors highlight persistent disparities in less developed regions and call for targeted prevention and treatment strategies to address these gaps.

Areas of improvement:

1. Limited Discussion of HICH-Specific Risk Factors:

While the manuscript identifies hypertension as the primary risk factor for HICH, it does not sufficiently explore other contributing factors (e.g., smoking, alcohol use, obesity) or their interactions with hypertension. The discussion of risk factors is brief and relies heavily on general statements.

Recommendation: Expand the discussion to include a detailed analysis of HICH-specific risk factors, supported by GBD data on attributable fractions or relevant literature. Discuss how these factors vary by region, sex, or SDI.

Your comments and suggestions are much appreciated and we have referred to the relevant literature to further analyze the specific risk factors for HICH and to further discuss how these factors differ by region, gender, or SDI. In the “Revised Manuscript with Track Changes”, lines 85-100.The additions are specified below:

In addition, factors such as hypertension, diet, and the environment can also have a significant impact on HICH risk across different regions, genders, and contexts such as the SDI[1].High systolic blood pressure is the most important factor for high ASDRs and age-standardized DALYs rates for ICH in 2021, followed by particulate matter pollution, cigarette smoking, indoor air pollution from solid fuels, and a high sodium diet, and there were gender differences in these risks, with the main factors influencing the risk of death for males being smoking and particulate matter pollution of the atmosphere, and the main factors influencing the risk of death for females being particulate matter pollution and indoor air pollution from solid fuels[2]. Among the different regions, Central and Southeast Asia have a higher prevalence of hypertension, which is coupled with unhealthy lifestyles such as high-salt diets and smoking, which further adds to the burden of ICH[3]. In low SDI areas, particulate matter pollution and smoking account for only 8.1% and 5.9% of all deaths from ICH, respectively, and household air pollution from solid fuels can account for 40.2% of all deaths. However, in areas with high SDI levels, deaths due to particulate matter pollution, smoking, and household air pollution are lower[2]

2. Inadequate Exploration of Data Limitations:

The manuscript acknowledges variability in data quality, particularly in low- and middle-income countries, but does not thoroughly discuss the implications of these limitations. For example, it does not address potential biases in HICH diagnosis, underreporting, or differences in healthcare access that may affect GBD estimates.

Recommendation: Dedicate a paragraph in the discussion to explore data limitations, including diagnostic accuracy, data completeness, and the ecological study design’s inability to infer causality. Discuss how these limitations may impact the findings and projections.

Thank you very much for your comments and suggestions, we have included a paragraph exploring the limitations of the data including diagnostic accuracy, data completeness, and the inability of ecological study designs to infer causality and discussing how these limitations may affect study results and predictions. In the “Revised Manuscript with Track Changes”, lines 526-561.The additions are specified below:

Strengths and limitations

The strength of this study lies in its systematic analysis of spatial and temporal trends in ASDRs and age-standardized DALYs rates for HICH by country, region, age, sex, and level of SDI using data from the GBD study from 1990 to 2021. We also projected the global burden of ASDR, age-standardized DALYs rates for HICH to 2050. To the best of our knowledge, this GBD-based study represents one of the more comprehensive efforts to date to analyze the global burden of HICH in terms of mortality rates, DALYs, and projected future trends; however, as described in previous studies[4-7], there are some limitations to the present study: Firstly, there may be challenges in diagnosing and reporting hypertensive cerebral hemorrhagic disease, i.e., variations in diagnostic criteria, reporting standards, and data collection methods in different countries and regions can affect the accuracy of results. Second, when GBD collects and organizes regional data, there may be incomplete data for some regions. For example, some studies may not provide regional data, while others may provide data that are inconsistent with GBD's regional delineation. In addition, the GBD database incorporates a relatively wide range of data, and the quality and availability of data vary significantly across countries and regions, especially in low- and middle-income countries that mostly lack robust data registries. As a result, the regional burden of HICH may not be accurately estimated, thereby affecting the effectiveness of targeted interventions and resource allocation. Third, GBD analyses rely on statistical models and assumptions to make inferences, which to some extent ignores real-world complexity factors and may not better reflect the lived experiences of people with HICH in different cultural contexts. Fourth, no risk factors for HICH were identified in the GBD database. HICH is a multifactorial disease, and its pathogenesis is related to genetics, environment, and lifestyle in addition to hypertension as a major factor. This study may not provide a comprehensive understanding of the risk factors for HICH, which to some extent hinders the development of effective prevention and control strategies. Finally, the impact of time should be considered. This analysis spanned a period of 30 years, and projections of the future burden of HICH were mostly based on current trends and patterns, during which major developments in medical technology, treatment protocols, and public health policies may have had an impact on the burden of HICH, which to a certain extent ignored the possible intervening factors that may have been relevant in the passage of time. Future research should focus on the impact of health policies, resource allocation, etc., and utilize multidisciplinary approaches and technologies to enhance the prevention, treatment, and global health outcomes of HICH.

3. Overreliance on Descriptive Statistics:

The results section is heavily descriptive, presenting extensive numerical data (e.g., ASDR, DALYs, EAPC) without sufficient synthesis or interpretation. This makes the section dense and challenging to follow for readers unfamiliar with GBD studies.

Recommendation: Streamline the results section by focusing on key trends and using tables/figures to summarize data. Provide more interpretive commentary to highlight the significance of findings (e.g., why certain regions have higher burdens).

Thank you very much for your comments and suggestions, we have streamlined the results section somewhat by focusing on the discussion of major trends and using tables/graphs to summarize the data, as well as providing more explanatory comments in the discussion section to emphasize the importance of the findings. In the “Revised Manuscript with Track Changes”, lines 121-364.Due to the large and mostly detailed issues involved, they are not listed separately here and have been labeled in the text.

4. Insufficient Detail on BAPC Model:

The manuscript briefly describes the BAPC model but does not explain its assumptions, limitations, or how it handles uncertainty in projections. Readers unfamiliar with this method may struggle to assess its validity.

Recommendation: Expand the methods section to include a detailed explanation of the BAPC model, including its mathematical framework, assumptions (e.g., stationarity of trends), and how it accounts for future changes in risk factors or healthcare access. Discuss potential limitations of long-term projections.

Thank you very much for your comments and suggestions, and we have expanded the Methods section to explain the BAPC model in detail, including its mathematical framework, assumptions, and how it takes into account future changes in risk factors or healthcare delivery, and to discuss potential limitations of long-term projections. In the “Revised Manuscript with Track Changes”, lines 161-177.The additions are specified below:

A Bayesian age-period-cohort (BAPC) model was employed to predict the ASDR, the DALYs, and the age-standardized DALYs rate for HICH from 2025 to 2050. BAPC modeling is a methodology used in epidemiology and biostatistics to analyze the relationship between incidence rates and time. It uses sample data and a priori information to obtain unique parameter estimates[8], allows for the inclusion of known risk factors as covariates in the model, and also simulates the impact of future changes in healthcare by setting up different scenarios. Based on the assumption that the effects of age, period, and cohort are similar in time, the Bayesian inference in the BAPC model utilizes second-order stochastic bias to smooth the three aforementioned prior values and predict the posterior rate[2].BAPC employs the Integrated Nested Laplace Approximation (INLA) to approximate the marginal posterior distributions, thereby avoiding the mixing and convergence problems associated with the Markov Chain Monte Carlo method and the traditional Bayesian approach , which has been widely used to analyze trends in chronic diseases and predict future disease burden[9]. Its long-term predictions are prone to deviate from the assumed conditions, and data quality and availability can suffer from the difficulty of incorporating unpredictable events.

5. Generic Policy Recommendations:

The policy recommendations (e.g., increasing screening, improving medical facilities) are broad and lack specificity regarding implementation, cost-effectiveness, or feasibility in low-resource settings. The manuscript does not address potential barriers, such as political or cultural resistance.

Recommendation: Provide more concrete policy recommendations, such as integrating HICH screening into existing hypertension programs, leveraging telemedicine for rural areas, or adopting cost-effective interventions like community health worker training. Discuss implementation challenges and potential solutions.

Your comments and suggestions are greatly appreciated, and we have addressed implementation challenges and potential solutions in the discussion section, as well as provided more specific policy recommendations, such as incorporating HICH screening into existing hypertension programs, utilizing telemedicine to serve rural areas, or cost-effective interventions such as community health worker training. In the “Revised Manuscript with Track Changes”, lines 461-464,562-590.The additions are specified below:

Strengthening medical infrastruct

---

## [Decision Letter · Decision Letter 1]

3 Jul 2025

Dear Dr. Ma,

Thank you for submitting your manuscript to PLOS ONE. After careful consideration, we feel that it has merit but does not fully meet PLOS ONE’s publication criteria as it currently stands. Therefore, we invite you to submit a revised version of the manuscript that addresses the points raised during the review process.

We look forward to receiving your revised manuscript.

Kind regards,

Thien Tan Tri Tai Truyen, M.D.

Academic Editor

PLOS ONE

Journal Requirements:

Reviewers' comments:

Reviewer's Responses to Questions

**Comments to the Author**

Reviewer #1: All comments have been addressed

Reviewer #2: All comments have been addressed

2. Is the manuscript technically sound, and do the data support the conclusions?

Reviewer #1: Yes

Reviewer #2: Yes

3. Has the statistical analysis been performed appropriately and rigorously?

Reviewer #1: Yes

Reviewer #2: Yes

4. Have the authors made all data underlying the findings in their manuscript fully available?

Reviewer #1: Yes

Reviewer #2: Yes

5. Is the manuscript presented in an intelligible fashion and written in standard English?

Reviewer #1: Yes

Reviewer #2: Yes

Reviewer #1: Good work. Authors have responded to reviewer comments appropriately. Global, regional, and national burden of hypertensive intracerebral hemorrhage, 1990 to 2021 and projections to 2050 is important to make changes in health policies considering the morbidity of hypertension in general population and its complications.

Reviewer #2: Thank you for submitting a revision and addressing the reviewer comments.

I have reviewed your manuscript, and I think that this paper is very important and would benefit and add to the current literature.

Abstract: In Lines 33–47, although the abstract is generally well-structured, these lines are slightly dense and long for an abstract; I suggest a clearer summarisation, for example, regional or country comparisons might be condensed and summarised as regional extremes in one concise sentence.

Intro: This is a very good and clear intor, however, it is subject for minor improvements. I found lines (78–91) very dense and lacking in flow and comprehensiveness. When I review papers, I usually suggest simplifying the meaning because if the paper is easy to read, it will be referenced a lot. Thus, I suggest that you summarise the key points and focus on their relevance to the HICH burden and disparities. Also, an important sentence to add is explaining why your paper is necessary despite previous GBD work. If you mention previous GBD work, what is different or important about your paper?

Methods: I think this section is very strong and defines how the paper has solid evidence. In line 121, insert a space after the URL. Otherwise, well done.

Results: This section is also well-written; however, I noticed that there were inconsistencies in terminology (for example, “countries” and “regions” are conflated); for instance, Southeast Asia is a region, not a country.

Discussion: This is a comprehensive and well-organised section. Still, there are a few areas that require refinement. Consider improving the transitions between paragraphs for a smoother flow, especially when shifting between global, regional, and national perspectives. In lines 343-344, referencing hormonal effects on neuroprotection is valuable; however, consider linking it to policy or clinical implications (e.g. gender-sensitive public health interventions). The final paragraph (lines 413-419) concludes well but could be enhanced by reiterating the practical importance of these findings, for example, how they might inform ongoing global health initiatives or SDG targets. The paragraph on limitations related to GBD data and methods (lines 427–447) is particularly strong; consider breaking it into two shorter paragraphs for improved readability. The Future Directions section is very strong; however, it would benefit from reordering for improved clarity. First outline research priorities, followed by the policy actions. For example, lines 457-461 (research) could precede lines 465–484 (policy).

**Do you want your identity to be public for this peer review?** For information about this choice, including consent withdrawal, please see our Privacy Policy

Reviewer #1: No

Reviewer #2: **Yes: ** Areej Almutairi

---

## [Author Response · Author response to Decision Letter 2]

6 Jul 2025

Response to Reviewers

Manuscript ID number:

PONE-D-25-19666

Title of paper:

Global, regional, and national burden of hypertensive intracerebral hemorrhage, 1990 to 2021 and projections to 2050: Results from the Global Burden of Disease Study 2021

Dear Editor and Reviewers,

We are again very grateful for the opportunity to revise the manuscript entitled “Global, regional, and national burden of hypertensive cerebral hemorrhage from 1990 to 2021 and projections to 2050: results from the Global Burden of Disease Study 2021” and would like to thank the reviewers again for their insightful comments. These comments were invaluable and very helpful in revising and refining the paper and provided important guidance for our research. In the submitted manuscript “Revised Manuscript with Track Changes”, we have retained all revision marks and provided detailed responses to the comments from the journal, reviewer 1 and reviewer 2, with the responses to reviewer 2 mainly marked in red. Below, we have provided a detailed response to each of the reviewers' comments and listed the number of lines of revision (for “Revised Manuscript with Track Changes” manuscripts) for ease of reference. In addition, we have completely revised the entire manuscript. In this response letter, reviewers' comments are italicized, and our corresponding revisions and additions to the manuscript are highlighted in red. We have endeavored to make all revisions clear and concise, and we hope that the revised manuscript will meet the requirements for publication.

Journal Requirements:

We have checked the reference list and this section is complete and correct and does not cite papers that have been retracted.

Response to reviewer #1:

Dear Reviewer #1,

Thank you very much for your comments on the manuscript. Thank you for your time and effort in reviewing the revised version of my paper.

Reviewer #1: Good work. Authors have responded to reviewer comments appropriately. Global, regional, and national burden of hypertensive intracerebral hemorrhage, 1990 to 2021 and projections to 2050 is important to make changes in health policies considering the morbidity of hypertension in general population and its complications.

Once again, thank you very much for your comments on the revised manuscript. We have thoroughly proofread the manuscript to ensure that it meets the requirements for publication in your journal.

Response to reviewer #2:

Dear Reviewer #2,

Thank you very much for your time and effort in providing valuable comments on our revised manuscript. We have revised the manuscript based on your feedback, and have elaborated on each of them below.

Thank you again for reviewing the manuscript, and I hope that my revisions will fulfill your requirements.

Reviewer #2: Thank you for submitting a revision and addressing the reviewer comments. I have reviewed your manuscript, and I think that this paper is very important and would benefit and add to the current literature.

Abstract:

In Lines 33–47, although the abstract is generally well-structured, these lines are slightly dense and long for an abstract; I suggest a clearer summarisation, for example, regional or country comparisons might be condensed and summarised as regional extremes in one concise sentence.

Thank you very much for your comments and suggestions. We have provided a clear and concise summary of the abstract, summarizing the extremes of the country and region in more concise sentences. In the “Revised Manuscript with Track Changes”, lines 47-57. The details of the modifications are as follows:

Country and regional patterns showed stark contrasts: Nauru and Mozambique had the highest ASDRs and age-standardized DALY rates, while Switzerland and Canada reported the lowest. Regionally, Central Africa, South Africa, Central Asia, East Asia, and Southeast Asia had the highest rates, whereas the Americas, Europe, and Oceania had the lowest. Age and gender trends indicated global peaks in the ASDRs (90–94 age group) and age-standardized DALY rates (85–89 age group), with men having higher rates across all age groups. Additionally, both ASDRs and age-standardized DALY rates were negatively associated with SDI levels. Projections from 2021 to 2050 suggest a continued overall decline in global ASDRs and age-standardized DALYs rates for HICH.

Introduction:

This is a very good and clear intor, however, it is subject for minor improvements. I found lines (78–91) very dense and lacking in flow and comprehensiveness. When I review papers, I usually suggest simplifying the meaning because if the paper is easy to read, it will be referenced a lot. Thus, I suggest that you summarise the key points and focus on their relevance to the HICH burden and disparities. Also, an important sentence to add is explaining why your paper is necessary despite previous GBD work. If you mention previous GBD work, what is different or important about your paper?

Thank you very much for your comments and suggestions. We have revised (78-91) to streamline the relevant exposition to make it more fluent, comprehensive, concise and easy to understand. In addition, we have added explanations of the necessity and importance of this thesis. In the “Revised Manuscript with Track Changes”, lines 101-110,123-132. The details of the modifications are as follows:

In 2021, high systolic blood pressure was a major contributor to high ASDRs and age-standardized DALYs. Other important risk factors include particulate matter pollution, smoking, indoor air pollution from solid fuels, and high sodium diets, and there are gender differences in these risks[1].Regionally, Central and Southeast Asia had a high prevalence of hypertension, exacerbated by unhealthy habits like high-salt diets and smoking, thus increasing the ICH burden[2]. In low SDI areas, household air pollution from solid fuels accounted for 40.2% of all ICH deaths, compared to only 8.1% and 5.9% for particulate matter pollution and smoking, respectively. In high SDI regions, deaths from these pollutants were lower[1].

Although most previous studies related to GBD [1] have focused on analyzing risk factors and distributional differences in ICH, this paper builds on previous studies by providing a detailed analysis of hypertension as an important risk factor and systematically analyzes the impact of geographic location, SDI, age, and sex differences on trends in ASDRs and age-standardized DALYs. To our knowledge, this GBD-based study is one of the most comprehensive studies to date analyzing the global burden of HICH in terms of mortality, DALYs, and projected future trends. These findings may inform public health interventions in specific regions, by gender, at different ages, etc., and may also provide valuable insights for the development of future prevention and management strategies.

Methods:

I think this section is very strong and defines how the paper has solid evidence. In line 121, insert a space after the URL. Otherwise, well done.

Again, thank you very much for your comment. We have inserted a space after the URL on line 121. In the “Revised Manuscript with Track Changes”, line 149.

Results:

This section is also well-written; however, I noticed that there were inconsistencies in terminology (for example, “countries” and “regions” are conflated); for instance, Southeast Asia is a region, not a country.

Again, thank you very much for commenting, we have double-checked this section for inconsistency and accuracy in wording. In the “Revised Manuscript with Track Changes”, lines 241-265.

Discussion:

This is a comprehensive and well-organised section. Still, there are a few areas that require refinement. Consider improving the transitions between paragraphs for a smoother flow, especially when shifting between global, regional, and national perspectives. In lines 343-344, referencing hormonal effects on neuroprotection is valuable; however, consider linking it to policy or clinical implications (e.g. gender-sensitive public health interventions). The final paragraph (lines 413-419) concludes well but could be enhanced by reiterating the practical importance of these findings, for example, how they might inform ongoing global health initiatives or SDG targets. The paragraph on limitations related to GBD data and methods (lines 427–447) is particularly strong; consider breaking it into two shorter paragraphs for improved readability. The Future Directions section is very strong; however, it would benefit from reordering for improved clarity. First outline research priorities, followed by the policy actions. For example, lines 457-461 (research) could precede lines 465–484 (policy).

Again, thank you very much for your comments. We have made the transition between global, regional and national perspectives more natural by adding transitional sentences and connecting words; we have linked hormonal effects on neuroprotection to public health interventions, In the “Revised Manuscript with Track Changes”, lines 370-376; and we have emphasized the importance of the study's findings for global health initiatives and the SDGs at the end to enhance the article's logic and usefulness, In the “Revised Manuscript with Track Changes”, lines 454-471.The details of the modifications are as follows:

Hormonal influences may explain some gender differences—estrogen’s neuroprotective effects in females delay ICH onset, underscoring the need for gender-sensitive interventions, such as tailored hypertension screening programs for men and women. Clinically, this could involve integrating sex-specific risk factor education into public health campaigns, particularly for smoking and air pollution exposure, which disproportionately affect males and females, respectively.

Our projections for the future indicate that the ASDR for HICH in 2050 will be 31.399 cases per 100,000 people, and the age-standardized DALYs rate will be 758.805 cases per 100,000 people. Although the global ASDR and age-standardized DALYs rate for HICH are on a downward trend, significant disparities persist across countries, regions, SDI levels, and genders. This prediction underscores the urgent need to prioritize targeted interventions in low-SDI and high-burden regions to align with Sustainable Development Goal 3 (SDG 3), which aims to reduce premature mortality from non-communicable diseases by one-third by 2030. For instance, the projected burden highlights the necessity of integrating HICH prevention into global health initiatives such as the World Health Organization’s Global Stroke Action Plan, particularly through initiatives like expanding hypertension screening in Central Sub-Saharan Africa and implementing gender-sensitive interventions to address male-female risk disparities. These findings also inform strategies for achieving universal health coverage (SDG 3.8), emphasizing the need to allocate resources toward healthcare infrastructure in regions with persistent high ASDRs, such as Southeast Asia and Oceania. By 2050, realizing these goals will require collaborative efforts to bridge care gaps, which our projections identify as critical for mitigating the ongoing global HICH burden.

Strengths and limitations:

Again, thank you very much for your comments. The paragraph on limitations of the GBD data and methodology (lines 427-447), we have split into two shorter paragraphs to improve readability. In the “Revised Manuscript with Track Changes”, lines 480-508.

Future directions

Again, thank you very much for your comments. We have reordered this section to make the exposition clearer. In the “Revised Manuscript with Track Changes”, lines 538-558. The details of the modifications are as follows:

Recommendations for future research and policy are essential to maintain and enhance HICH disease control. First, further research should focus on understanding the underlying causes of ASDR and age - standardized DALYs rate anomalies in regions like sub-Saharan Africa, examining socio-cultural, economic, and environmental factors, such as how local cultures, religious beliefs, and economic development influence HICH morbidity and mortality. Second, exploring the interactive effects of genetics, diet, and pollution on HICH risk across different SDI levels can provide a basis for formulating targeted prevention strategies. Third, evaluating the impact of emerging technologies, such as telemedicine and AI-based diagnostic tools, on HICH management in resource-constrained areas is crucial for optimizing care delivery.

Policy actions must build on these research directions to enhance HICH control. Low- and middle-SDI regions need to strengthen healthcare infrastructure, which includes integrating HICH screening into existing hypertension programs, training community health workers, and establishing telemedicine networks to serve rural areas. Policy initiatives should also prioritize formulating national HICH prevention plans and integrating them into public health strategies, expanding health insurance coverage for low-income groups and increasing the reimbursement for HICH-related medical expenses, promoting international cooperation to share prevention technologies and best practices, and implementing socio-economic development programs such as air pollution control, the provision of safe drinking water, and the construction of public fitness facilities to address upstream risk factors.

Again, thank you very much for your comments on my paper. I appreciate your feedback and suggestions. I will revise my paper according to your comments and submit a revised version soon. I hope you will find the revised paper satisfactory and acceptable for publication. Thank you for your time and effort.

Sincerely, [Chao Zhang]

Reference

1.Song D, Xu D, Li M, Wang F, Feng M, Badr A, et al. Global, regional, and national burdens of intracerebral hemorrhage and its risk factors from 1990 to 2021. Eur J Neurol. 2025;32(1):e70031. Epub 2024/12/28. doi: 10.1111/ene.70031. PubMed PMID: 39731311; PubMed Central PMCID: PMCPMC11680743.

2.Schutte AE, Srinivasapura Venkateshmurthy N, Mohan S, Prabhakaran D. Hypertension in Low- and Middle-Income Countries. Circ Res. 2021;128(7):808-26. Epub 2021/04/02. doi: 10.1161/circresaha.120.318729. PubMed PMID: 33793340; PubMed Central PMCID: PMCPMC8091106.

---

## [Decision Letter · Decision Letter 2]

18 Jul 2025

Dear Dr. Ma,

Thank you for submitting your manuscript to PLOS ONE. After careful consideration, we feel that it has merit but does not fully meet PLOS ONE’s publication criteria as it currently stands. Therefore, we invite you to submit a revised version of the manuscript that addresses the points raised during the review process.

**ACADEMIC EDITOR: Additional comments from reviewers are valid. I believe an additional revision based on those comments is appropriate. **

We look forward to receiving your revised manuscript.

Kind regards,

Thien Tan Tri Tai Truyen, M.D.

Academic Editor

PLOS ONE

Journal Requirements:

Reviewers' comments:

Reviewer's Responses to Questions

**Comments to the Author**

Reviewer #1: All comments have been addressed

Reviewer #2: All comments have been addressed

2. Is the manuscript technically sound, and do the data support the conclusions?

Reviewer #1: Yes

Reviewer #2: Yes

3. Has the statistical analysis been performed appropriately and rigorously?

Reviewer #1: Yes

Reviewer #2: Yes

4. Have the authors made all data underlying the findings in their manuscript fully available?

Reviewer #1: Yes

Reviewer #2: Yes

5. Is the manuscript presented in an intelligible fashion and written in standard English?

Reviewer #1: Yes

Reviewer #2: Yes

Reviewer #1: Good work. This article will help the literature on national burden of hypertensive intracerebral hemorrhage, 1990 to 2021 and projections to 2050.

Reviewer #2: Dear Author, Thank you for choosing PLOS ONE and submitting your manuscript titled ‘Global, regional, and national burden of hypertensive intracerebral haemorrhage, 1990 to 2021 and projections to 2050: Results from the Global Burden of Disease Study 2021’. This is an important topic that adds to the current literature. Although I think this paper is good, I believe that there are minor improvements that could be addressed.

Introduction: The introduction addresses the research question well and provides a good reference to the literature on global and local contexts. In lines 60-62 the sentence ‘.The sentence Hypertensive intracerebral haemorrhage (HICH) is the predominant… contains grammatical errors. Although the information in lines Lines 64-73, althoug they is correct, the transition is somewhat confusing; consider writing ‘Despite improvements in clinical management, disparities remain...” to enhance the flow of the section. In reference 8, I think the figures provided require further clarification. I recommend specifying the populations studied or the source of these percentages for context because it is vague, and I am not sure how these figures are important or can be relied on. In lines 87-91,

I noticed that the paragraph reiterates several ideas already mentioned earlier (e.g. high systolic blood pressure, regional variation, and SDI). I suggest condensing these points to avoid redundancy and to maintain reader engagement. also, When you mentioned “to bridge this gap,” I think it would be clearer to say exactly what gap they aim to address—such as “the lack of data on predictors of stroke length of stay in Indonesian hospitals.” Finally, I recommend refining the last sentence to be more specific and precise. Rather than stating that it will help “health service planning and management”, perhaps stating the improvement of discharge protocols or resource allocation would be better and more focused.

Methods: This section is also well written and subject to minor suggestions. I suggest clarifying the data source in line 107 of the manuscript. It is not obvious how the February 2025 collection period applies when using retrospective GBD data. This could confuse the readers. In addition, I believe it is worth rephrasing line 114. Registering in a "paper proposal form" is vague. Do the authors mean the GBD disease proposal database? Clarify its purpose. The definition of HICH in the GBD dataset is not mentioned; therefore, I recommend briefly reminding readers how HICH was defined, even if it was described in prior publications. In lines 122-123, the reference to “prior publications” is too vague. We have named one major methodological reference and summarised the key methods used. The information in lines 127–135 is repeated the same info twice. I suggest condensing to avoid redundancies. I felt confused reading this section because I am not sure why stratification is important in the context of HICH. I recommend providing a brief explanation. In lines 154–156, the authors introduce limitations ("long-term predictions deviate…"), but I think this should be moved to the discussion or limitations section instead of being buried here.

Results: I think this section is comprehensive and data-rich. The figures and trends are clearly presented. In line 199, the phrase “a larger downward trend” is vague and could be supported by EAPC values to quantify the differences more clearly. Beside that, good job.

Discussion: I think this is a well-structured and data-rich paper. I suggest rephrasing line 321 to emphasize the novelty more clearly for example, “This is the first study to explore HICH burden using updated 2021 GBD data stratified by SDI, age, and sex.” I suggest rephrasing line 321 to emphasize the novelty more clearly for example, “This is the first study to explore HICH burden using updated 2021 GBD data stratified by SDI, age, and sex.” In lines 333-336, the explanation for sex-based trends is informative. However, I suggest clarifying that the differences in male and female trends may also reflect differences in healthcare access or behaviour, not just biology. In lines 349-351, I would suggest interpreting why Nauru and Mozambique show such high values as outliers due to data quality, small population size, or actual health system deficiencies. In line 403, I recommend adding 1-2 words on “how” oestrogen lowers lipid levels and affects vasodilation to strengthen biological plausibility. In addition, in line 417, I recommend briefly stating what happened during the “reversal” in 1999–2003—was it due to war, policy changes, or data artefacts? in lines 471-472, I think a concrete suggestion, such as using electronic health records or AI for surveillance, would enhance the future directions.

**Do you want your identity to be public for this peer review?** For information about this choice, including consent withdrawal, please see our Privacy Policy

Reviewer #1: No

Reviewer #2: **Yes: ** Areej Almutairi

---

## [Author Response · Author response to Decision Letter 3]

22 Jul 2025

Response to Reviewers

Manuscript ID number:

PONE-D-25-19666R2

Title of paper:

Global, regional, and national burden of hypertensive intracerebral hemorrhage, 1990 to 2021 and projections to 2050: Results from the Global Burden of Disease Study 2021

Dear Editor and Reviewers,

We are again very grateful for the opportunity to revise the manuscript entitled “Global, regional, and national burden of hypertensive cerebral hemorrhage from 1990 to 2021 and projections to 2050: results from the Global Burden of Disease Study 2021” and would like to thank the reviewers again for their insightful comments. These comments were invaluable and very helpful in revising and refining the paper and provided important guidance for our research. In the submitted manuscript “Revised Manuscript with Track Changes”, we have retained all revision marks and provided detailed responses to the comments from the journal, reviewer 1 and reviewer 2, with the responses to reviewer 2 mainly marked in red. Below, we have provided a detailed response to each of the reviewers' comments and listed the number of lines of revision (for “Revised Manuscript with Track Changes” manuscripts) for ease of reference. In addition, we have completely revised the entire manuscript. In this response letter, reviewers' comments are italicized, and our corresponding revisions and additions to the manuscript are highlighted in red. We have endeavored to make all revisions clear and concise, and we hope that the revised manuscript will meet the requirements for publication.

Journal Requirements:

1.If the reviewer comments include a recommendation to cite specific previously published works, please review and evaluate these publications to determine whether they are relevant and should be cited. There is no requirement to cite these works unless the editor has indicated otherwise.

We have reviewed and evaluated the previously published works cited to ensure that they are relevant to what is being discussed.

2.Please review your reference list to ensure that it is complete and correct. If you have cited papers that have been retracted, please include the rationale for doing so in the manuscript text, or remove these references and replace them with relevant current references. Any changes to the reference list should be mentioned in the rebuttal letter that accompanies your revised manuscript. If you need to cite a retracted article, indicate the article’s retracted status in the References list and also include a citation and full reference for the retraction notice.

We have checked the reference list and this section is complete and correct and does not cite papers that have been retracted.

Response to reviewer #1:

Dear Reviewer #1,

Thank you very much for your comments on the manuscript. Thank you for your time and effort in reviewing the revised version of my paper.

Reviewer #1: Good work. This article will help the literature on national burden of hypertensive intracerebral hemorrhage, 1990 to 2021 and projections to 2050.

Once again, thank you very much for your comments on the revised manuscript. We have thoroughly proofread the manuscript to ensure that it meets the requirements for publication in your journal.

Response to reviewer #2:

Dear Reviewer #2,

Thank you very much for your time and effort in providing valuable comments on our revised manuscript. We have revised the manuscript based on your feedback, and have elaborated on each of them below.

Thank you again for reviewing the manuscript, and I hope that my revisions will fulfill your requirements.

Reviewer #2: Dear Author, thank you for choosing PLOS ONE and submitting your manuscript titled ‘Global, regional, and national burden of hypertensive intracerebral haemorrhage, 1990 to 2021 and projections to 2050: Results from the Global Burden of Disease Study 2021’. This is an important topic that adds to the current literature. Although I think this paper is good, I believe that there are minor improvements that could be addressed.

Introduction:

1.The introduction addresses the research question well and provides a good reference to the literature on global and local contexts. In lines 60-62 the sentence ‘.The sentence Hypertensive intracerebral haemorrhage (HICH) is the predominant… contains grammatical errors.

Thank you very much for your comments and suggestions. We have corrected lines 60-62 where it says "Hypertensive cerebral hemorrhage (HICH) is the main ...... " grammatical error. In the “Revised Manuscript with Track Changes”, lines 60-62. The details of the modifications are as follows:

Hypertensive intracerebral hemorrhage (HICH) is the predominant subtype of ICH, is strongly associated with hypertension�and accounts for approximately 70% of all ICH cases.

2.Although the information in lines Lines 64-73, althoug they is correct, the transition is somewhat confusing; consider writing ‘Despite improvements in clinical management, disparities remain...” to enhance the flow of the section.

We have revised the transition after lines 64-73 to “Despite improvements in clinical management, disparities remain ......” to enhance the flow of the section. In the “Revised Manuscript with Track Changes”, lines 77-84. The details of the modifications are as follows:

Despite improvements in clinical management, high systolic blood pressure, a major risk factor for HICH, continues to be strongly associated with factors such as smoking, diet, and the environment, and varies by country, region, gender, and SDI level[1].

3.In reference 8, I think the figures provided require further clarification. I recommend specifying the populations studied or the source of these percentages for context because it is vague, and I am not sure how these figures are important or can be relied on.

We have further clarified the numbers provided in Reference 8 to be more specific about the populations studied or the source of the percentages, thus improving the credibility of the data. In the “Revised Manuscript with Track Changes”, lines 67-71. The details of the modifications are as follows:

Studies on the treatment of ICH have found that blood pressure control trials reduce ICH mortality by 10-15%[2], after administration of anticoagulant-specific reversal agents (e.g., idarucizumab, prothrombin complex concentrate), mortality and hematoma enlargement rates in patients with ICH are reduced by approximately 30%-40% compared to baseline levels without reversal agents or conventional therapy[3].

4.In lines 87-91, I noticed that the paragraph reiterates several ideas already mentioned earlier (e.g. high systolic blood pressure, regional variation, and SDI). I suggest condensing these points to avoid redundancy and to maintain reader engagement. also, when you mentioned “to bridge this gap,” I think it would be clearer to say exactly what gap they aim to address—such as “the lack of data on predictors of stroke length of stay in Indonesian hospitals.”

We have streamlined lines 87-91 where there is duplication from before. In addition, we further elaborated on what should be “to bridge this gap”. In the “Revised Manuscript with Track Changes”, lines 100-106. The details of the modifications are as follows:

Therefore, there is a need to explore why differences in spatial and temporal trends, gender, and age exist in HICH—for example, "in the Republic of Nauru, patients have low income levels and do not have access to standardized treatments[4], which can complicate and make hospitalization unpredictable; in the Republic of Mozambique, there is a severe shortage of health workers[5], which can prevent timely treatment and prolong hospitalization"—to guide prevention and treatment efforts.

5.Finally, I recommend refining the last sentence to be more specific and precise. Rather than stating that it will help “health service planning and management”, perhaps stating the improvement of discharge protocols or resource allocation would be better and more focused.

We have amended the last sentence to read “it could help to improve discharge protocols or resource allocation”. In the “Revised Manuscript with Track Changes”, lines 116-118. The details of the modifications are as follows:

These findings may inform public health interventions in specific regions, by gender, at different ages, etc., and can also help improve discharge protocols or resource allocation.

Methods:

1.This section is also well written and subject to minor suggestions. I suggest clarifying the data source in line 107 of the manuscript. It is not obvious how the February 2025 collection period applies when using retrospective GBD data. This could confuse the readers.

Again, thank you very much for your comment. We have elaborated on the data sources in line 107 and explained how the use of retrospective GBD data relates to the February 2025 collection period. In the “Revised Manuscript with Track Changes”, lines 121-125.The details of the modifications are as follows:

The data utilized in this study were derived from publicly available retrospective data from the GBD study. For the purpose of data extraction and collation in this manuscript, the specific operational period for accessing and organizing these GBD data was from February 5, 2025, to February 16, 2025.

2.In addition, I believe it is worth rephrasing line 114. Registering in a "paper proposal form" is vague. Do the authors mean the GBD disease proposal database? Clarify its purpose. The definition of HICH in the GBD dataset is not mentioned; therefore, I recommend briefly reminding readers how HICH was defined, even if it was described in prior publications.

The platform contacted on line 114 (https://uwhealthmetrics.co1.qualtrics.com/) is a questionnaire platform that is not directly linked to the GBD database and is the channel used to collate the data. Our data source is the Global Health Data Exchange query tool (GHDx) (http://ghdx.healthdata.org/gbd-results-tool). In addition, we describe the definition of HICH. In the “Revised Manuscript with Track Changes”, lines 133-137.The details of the modifications are as follows:

HICH is usually a spontaneous intracerebral hemorrhage that occurs in patients with a history of chronic hypertension (blood pressure ≥140/90 mmHg or a previous diagnosis of hypertension) and excludes cases caused by other etiologies (e.g., cerebral aneurysms, arteriovenous malformations, trauma, or coagulation disorders) [6].

3.In lines 122-123, the reference to “prior publications” is too vague. We have named one major methodological reference and summarised the key methods used.

We have been more specific about “prior publications” in lines 122-123.In the “Revised Manuscript with Track Changes”, lines 143-152.The details of the modifications are as follows:

The data from the GBD2021 study were processed and analyzed as follows:Data were statistically analyzed and visualized using R 4.4.1 software. A linear regression model was used to calculate the EAPC and its 95% CI to assess the temporal trends of age-standardized rates (ASRs). A BAPC model was used to approximate marginal posterior distributions using the Integrated Nested Laplace Approximation (INLA) method to project ASDRs and age-standardized DALYs rates for HICH from 2025 to 2050[7,8].

4.The information in lines 127–135 is repeated the same info twice. I suggest condensing to avoid redundancies. I felt confused reading this section because I am not sure why stratification is important in the context of HICH. I recommend providing a brief explanation.

We have removed and streamlined the repetitive information in lines 127-135 and explained the important role of SDI layering in HICH. In the “Revised Manuscript with Track Changes”, lines 168-171.The details of the modifications are as follows:

SDI stratification is critical to HICH research. It takes into account differences in health care access, prevention, and management across SDI levels and facilitates targeted analysis of socio-economic influences on HICH trends, thus helping to target interventions to specific regions.

5.In lines 154–156, the authors introduce limitations ("long-term predictions deviate…"), but I think this should be moved to the discussion or limitations section instead of being buried here.

We have deleted (“long-term predictions deviate...”) in lines 154-156 and moved it to the limitations section. In the “Revised Manuscript with Track Changes”, lines 518-521.

Results:

I think this section is comprehensive and data-rich. The figures and trends are clearly presented. In line 199, the phrase “a larger downward trend” is vague and could be supported by EAPC values to quantify the differences more clearly. Beside that, good job.

We apologize that we did not find the “larger downward trend” in the Results section after a careful search in and around line 199, which may be due to an oversight in our understanding of the comment. In order to improve the manuscript, I would like to ask you to point out the specific place where the change is needed, and I will immediately make the change according to your suggestion. I am deeply sorry for the inconvenience caused to you, and thank you again for your patience and guidance!

Discussion:

1.I think this is a well-structured and data-rich paper. I suggest rephrasing line 321 to emphasize the novelty more clearly for example, “This is the first study to explore HICH burden using updated 2021 GBD data stratified by SDI, age, and sex.” I suggest rephrasing line 321 to emphasize the novelty more clearly for example, “This is the first study to explore HICH burden using updated 2021 GBD data stratified by SDI, age, and sex.”

Again, thank you very much for your comments. We have revised the wording of line 321 to more clearly emphasize novelty. In the “Revised Manuscript with Track Changes”, lines 353-359. The details of the modifications are as follows:

This is the first study to explore HICH burden using updated 2021 GBD data stratified by SDI, age, and sex, and the findings have important value for the development of global health policy and the effective allocation of resources.

2.In lines 333-336, the explanation for sex-based trends is informative. However, I suggest clarifying that the differences in male and female trends may also reflect differences in healthcare access or behaviour, not just biology.

We have revised the interpretation of rows 333-336 based on gender trends to further clarify that differences in male and female trends may also reflect differences in health care access or behavior, not just biological differences. In the “Revised Manuscript with Track Changes”, lines 378-395. The details of the modifications are as follows:

In addition to biological factors, differences in gender trends are also reflected in access to health care and daily behaviors. Relevant studies have found that among hospital admissions, a higher proportion of men (85.6%) than women (74.7%) were admitted to acute stroke units or acute neurological intensive care units[9], mostly due to men's busy schedules, low health awareness, and failure to seek medical attention in a timely manner, which resulted in more serious conditions. Smoking and excessive alcohol consumption are important risk factors for stroke, with men having more current smoking (23.3% vs. 4.2%) and excessive alcohol consumption (16.5% vs. 1.3%) behaviors than women, which in turn increases the risk of stroke[10]. This highlights the need for gender-sensitive interventions, such as tailoring hypertension screening programs for men and women. Clinically, this may require incorporating gender-specific risk factor education into public health campaigns, especially for men with low health awareness and smoking and excessive drinking behaviors.

3.In lines 349-351, I would suggest interpreting why Nauru and Mozambique show such high values as outliers due to data quality, small population size, or actual health system deficiencies.

We have provided an explanation of why Nauru and Mozambique show such high values in rows 349-351.In the “Revised Manuscript with Track Changes”, lines 403-409. The details of the modifications are as follows:

It is worth noting that the abnormally high values observed in Nauru and Mozambique may be due to limited data qual

---

## [Editor Report · Decision Letter 3]

28 Oct 2025

Global, regional, and national burden of hypertensive intracerebral hemorrhage, 1990 to 2021 and projections to 2050: Results from the Global Burden of Disease Study 2021

PONE-D-25-19666R3

Dear Dr. Ma,

We’re pleased to inform you that your manuscript has been judged scientifically suitable for publication and will be formally accepted for publication once it meets all outstanding technical requirements.

Kind regards,

Atakan Orscelik

Academic Editor

PLOS ONE
---

## [Editor Report · Acceptance letter]

PONE-D-25-19666R3

PLOS ONE

Dear Dr. Ma,

I'm pleased to inform you that your manuscript has been deemed suitable for publication in PLOS ONE. Congratulations! Your manuscript is now being handed over to our production team.

Kind regards,

on behalf of

Dr. Atakan Orscelik

Academic Editor

PLOS ONE